# Greenhouse gas observations from the Northeast Corridor tower network

Anna Karion[1], William Callahan[2], Michael Stock[2], Steve Prinzivalli[2], Kristal R. Verhulst[3], Jooil Kim[4], Peter K. Salameh[4], Israel Lopez-Coto[5], James Whetstone[1]

[1]Special Programs Office, National Institute of Standards and Technology, Gaithersburg, MD, USA
[2]Earth Networks, Inc., Germantown, MD, USA
[3]NASA Jet Propulsion Laboratory, California Institute of Technology, Pasadena, CA, USA
[4]Scripps Institution of Oceanography, University of California San Diego, La Jolla, CA, USA
[5]Engineering Laboratory, National Institute of Standards and Technology, Gaithersburg, MD, USA

*Correspondence to*: Anna Karion (Anna.Karion@nist.gov)

**Abstract.** We present the organization, structure, instrumentation, and measurements of the Northeast Corridor greenhouse gas observation network. This network of tower-based in-situ carbon dioxide and methane observation stations was established in 2015 with the goal of quantifying emissions of these gases in urban areas in the north-eastern United States. A specific focus of the network is the cities of Baltimore, Maryland, and Washington, D.C., USA, with a high density of observation stations in these two urban areas. Additional observation stations are scattered throughout the northeast US, established to complement other existing urban and regional networks and to investigate emissions throughout this complex region with a high population density and multiple metropolitan areas. Data described in this paper are archived at the National Institute of Standards and Technology and can be found at https://doi.org/10.18434/M32126 (Karion et al., 2019).

## 1 Introduction

As the population of cities grows globally due to trends toward urbanization, so does their relative contribution to global anthropogenic greenhouse gas (GHG) budgets (Edenhofer O., 2014; O'Neill et al., 2010). City governments are making commitments to reduce their emissions of GHGs through various sustainability and efficiency measures and coordination with organizations like the C40 Climate Leadership Group (www.c40.org), the Global Covenant of Mayors for Climate and Energy (www.globalcovenantofmayors.org), and others. These organizations require individual cities to conform to certain standardized mechanisms and practices for reporting their carbon emissions. City governments rely on inventories compiled using data on fuel use, energy usage, etc. to estimate their total emissions and changes over time, and to determine the efficacy of various emissions mitigation policies. Analysis of atmospheric measurements provides additional useful information to such efforts, by confirming inventory estimates (Sargent et al., 2018; Lauvaux et al., 2016), detecting trends (Mitchell et al., 2018), or estimating emissions that are not well quantified using inventory methods, such as methane emissions (McKain et al., 2015; Ren et al., 2018; Lamb et al., 2016; Yadav et al., 2019). Several urban top-down measurement efforts are underway in various

cities that include networks of observations, often in-situ $CO_2$ and $CH_4$ measurements from rooftops or towers (Verhulst et al., 2017; Xueref-Remy et al., 2018; Bares et al., 2019), or using other long-path and remote sensing methods (Waxman et al., 2019; Hedelius et al., 2018; Wong et al., 2016; Pillai et al., 2016).


The National Institute of Standards and Technology (NIST) has partnered with other federal, private, and academic institutions to establish three urban testbeds in the United States: the Indianapolis Flux Experiment (INFLUX, influx.psu.edu), the Los Angeles Megacities Carbon Project (megacities.jpl.nasa.gov), and the Northeast Corridor (NEC, www.nist.gov/topics/northeast-corridor-urban-test-bed). The goals of the urban testbeds are to develop and refine techniques

for estimating greenhouse gas emissions from cities and to understand the uncertainty of emissions estimates at various spatial and temporal scales (e.g., whole city annual emissions vs. 1-km weekly emissions). Recent results from the longest-running testbed, INFLUX, show that whole city emissions can be estimated using three different methods to within 7% (Turnbull et al., 2019).

The Northeast Corridor (NEC) was established in 2015 as the third NIST urban testbed. The goals for this project are to demonstrate that top-down atmospheric emissions estimation methods can be used in a domain that is complicated by many upwind and nearby emissions sources in the form of surrounding urban areas. The objective is to isolate the anthropogenic GHG emissions from urban areas along the U.S. East Coast from many confounding sources upwind (cities, oil and gas development, coal mines, and power plants) and from the large biological $CO_2$ signal from the highly productive forests nearby

and within the cities. The presence of highly vegetated areas such as urban parks, local agriculture, and managed lawns is expected to dominate the $CO_2$ signal in summertime, as has been found in Boston, Massachusetts (Sargent et al., 2018). The NEC project has a current focus on the urban areas of Washington, D.C. and Baltimore, Maryland, U.S.A., with existing plans to expand northward to cover the entire urbanized corridor of the northeast U.S., including the cities of Philadelphia and New York City, and eventually linking up with existing measurement stations in Boston, Massachusetts (McKain et al., 2015;

Sargent et al., 2018).

The NEC project includes multiple measurement and analysis components. The backbone of the NEC project is a network of in-situ $CO_2$ and $CH_4$ observation stations with continuous high-accuracy measurements of these two greenhouse gases. In addition, periodic flight campaigns of multiple weeks each year are conducted by the University of Maryland (FLAGG-MD, www.atmos.umd.edu/~flaggmd) and Purdue University (https://www.science.purdue.edu/shepson/research/ALARGreenhouseGas/), focusing on wintertime observations of $CO_2$, $CH_4$, CO, $O_3$, $SO_2$, $NO_2$, from instrumented aircraft (Ren et al., 2018; Salmon et al., 2018; Lopez-Coto et al., 2020). The use of low-cost $CO_2$ sensors is also being investigated in Washington, D.C., with work focusing on calibration and determination of long-term stability of inexpensive non-dispersive infrared (NDIR) sensors with potential for use in $CO_2$ data assimilation

techniques (Martin et al., 2017). The NEC project also includes an extensive modelling component. First, high-resolution

meteorological modelling (using the Weather Research and Forecast (WRF) model) is being conducted (Lopez-Coto et al., 2019), with output coupled to Lagrangian dispersion models such as STILT (Lin et al., 2003; Nehrkorn et al., 2010) and HYSPLIT (Stein et al., 2015). These transport and dispersion models are used to interpret observations from both aircraft and tower stations and in atmospheric inverse analyses to estimate fluxes of $CO_2$ and $CH_4$ from the cities of Washington, D.C. and Baltimore, Maryland (Lopez-Coto et al., 2020; Huang et al., 2019). A high-resolution fossil-fuel $CO_2$ inventory, Hestia, is also being developed for this project (Gurney et al., 2012; Gurney et al., 2019).

Here we focus on the high-accuracy tower observation network and associated data collection and processing methods. Section 2 describes the tower network design and characterizes the different site locations; Section 3 describes the measurement methods, instrumentation, and calibration; Section 4 presents the uncertainty derivation for the measurements; and finally, Section 5 presents some of the observations from the current record.

## 2 Network design and site characterization

The NEC project includes 29 observation stations, all managed and operated by Earth Networks, Inc [1] . (www.earthnetworks.com/why-us/networks/greenhouse-gas). Ten stations were existing Earth Networks (EN) measurement sites in the northeast U.S. that became part of the NEC project in 2015. Nineteen stations were established (or will be established) specifically for the NEC project, with site locations identified by NIST. Sixteen of these station locations were chosen to be used for emissions estimation in a domain around Baltimore and Washington, D.C. (red boundary, Fig. 1) using inverse modelling techniques (Lopez-Coto et al., 2017; Mueller et al., 2018). Three others are in Mashpee, MA, Philadelphia, PA, and Waterford Works, NJ. As of publication, 14 of these 19 have been established, with delays occurring due to difficulty finding suitable tower locations to agree to house the systems. The hardware and software operating all the sites is identical with few exceptions as noted in the text.

---

[1] Certain commercial equipment, instruments, or materials are identified in this paper in order to specify the experimental procedure adequately. Such identification is not intended to imply recommendation or endorsement by the National Institute of Standards and Technology, nor is it intended to imply that the materials or equipment identified are necessarily the best available for the purpose.

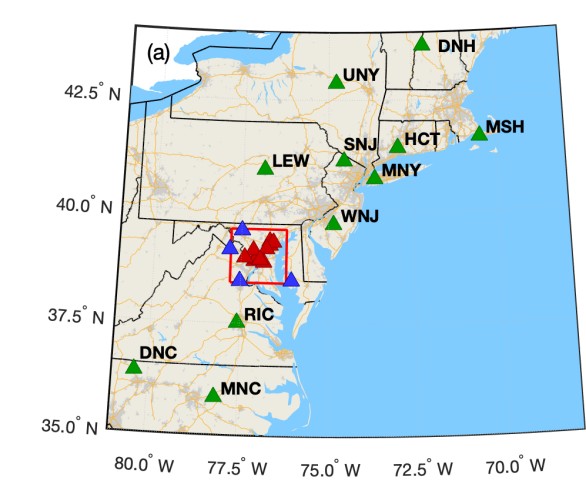 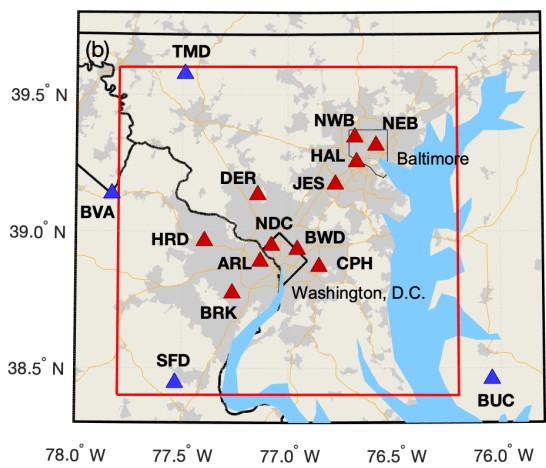

**Figure 1. Locations of Northeast Corridor (NEC) established tower-based observations, corresponding to Table 1. The red rectangle indicates the modelling analysis domain. Light grey shading indicates census-designated urban areas; yellow lines are interstate highways; black boundaries are state lines, with a thinner black line showing the City of Baltimore. Green triangles indicate regional sites, red triangles indicate urban sites, while blue triangles are more rural or background sites surrounding the Washington/Baltimore domain. (a) Regional map. (b) Inset focusing on Washington, D.C., and Baltimore, Maryland. All map data layers obtained from either Natural Earth (naturalearthdata.com) or U.S. Government sources (www.census.gov) and are in the public domain.**

The initial design of the core urban Baltimore/Washington network was focused on optimizing tower site locations with the goal of reducing uncertainty in estimating anthropogenic $CO_2$ emissions from Washington, D.C., and Baltimore using an atmospheric inversion model (Lopez-Coto et al., 2017). Twelve communications towers were identified as part of that study as ideal locations for measurements. Actual measurement sites were sometimes established at locations near the ideal study location usually due to logistical difficulties obtaining leases at the ideal tower sites. A second design study determined ideal locations for background stations, i.e. observation station locations that would aid in the determination of background $CO_2$ entering the analysis domain (Mueller et al., 2018). Four stations were identified as part of that study; an existing EN site in Bucktown, MD, serves as a fifth background station southeast of the analysis domain (Fig. 1). Although inlet heights were desired to be 100 m above ground level (agl), often shorter towers were used due to the lack of availability of tall towers in ideal locations; the shortest tower in this network has the uppermost inlet at 38 m agl (HRD). Table 1 indicates details and locations of each site.

**Table 1. Northeast Corridor Site List. Sites lacking location information are still in the planning phase, with no assigned site code or exact location. *If a station was established prior to the beginning of the project, its established date is listed as 1/2015, the start date of the project. Data prior to this date is not part of the NIST data release.**

| Site Code | EN Site ID | Location | Latitude | Longitude | Elev. (m) | Intake Heights (m) | Measure-ments | Est. (mo./yr.) |
|---|---|---|---|---|---|---|---|---|
| **Northeast Corridor Regional Sites (12)** | | | | | | | | |
| DNC | GHG12 | Danbury, NC | 36.3769 | -80.3689 | 703 | 100/50 | $CO_2$, $CH_4$ | 1/2015* |
| MNC | GHG15 | Middlesex, NC | 35.8313 | -78.1453 | 74 | 213/50 | $CO_2$, $CH_4$ | 1/2015* |
| RIC | GHG18 | Richmond, VA | 37.5088 | -77.5761 | 104 | 95/50 | $CO_2$, $CH_4$ | 1/2015* |
| SNJ | GHG19 | Stockholm, NJ | 41.1436 | -74.5387 | 406 | 53/42 | $CO_2$, $CH_4$ | 1/2015* |
| HCT | GHG21 | Hamden, CT | 41.4337 | -72.9452 | 204 | 100/50 | $CO_2$, $CH_4$ | 1/2015* |
| LEW | GHG25 | Lewisburg, PA | 40.9446 | -76.8789 | 166 | 95/50 | $CO_2$, $CH_4$ | 1/2015* |
| DNH | GHG35 | Durham, NH | 43.7089 | -72.1541 | 559 | 100/50 | $CO_2$, $CH_4$ | 1/2015* |
| UNY | GHG38 | Utica, NY | 42.8790 | -74.7852 | 483 | 45/35 | $CO_2$, $CH_4$ | 1/2015* |
| MNY | GHG47 | Mineola, NY | 40.7495 | -73.6384 | 34 | 90/50 | $CO_2$, $CH_4$ | 1/2015* |
| MSH | GHG54 | Mashpee, MA | 41.6567 | -70.4975 | 32 | 46/25 | $CO_2$, $CH_4$, CO | 12/2015 |
| WNJ | GHG69 | Waterford Works, NJ | 39.7288 | -74.8441 | 33 | 241/201/151/98/43 | $CO_2$, $CH_4$ | Planned 2020 |
| | | Philadelphia, PA | | | | | $CO_2$, $CH_4$ | |
| **Washington, D.C. and Baltimore Urban Sites (12)** | | | | | | | | |
| HAL | GHG48 | Halethorpe, MD | 39.2552 | -76.6753 | 70 | 58/29 | $CO_2$, $CH_4$ | 10/2015 |
| ARL | GHG55 | Arlington, VA | 38.8917 | -77.1317 | 111 | 92/50 | $CO_2$, $CH_4$ | 1/2016 |
| NDC | GHG56 | Northwest DC | 38.9499 | -77.0796 | 128 | 91/45 | $CO_2$, $CH_4$ | 12/2015 |
| NWB | GHG58 | NW Baltimore, MD | 39.3445 | -76.6851 | 135 | 55/27 | $CO_2$, $CH_4$ | 9/2016 |
| NEB | GHG59 | NE Baltimore, MD | 39.3154 | -76.5830 | 44 | 67/50 | $CO_2$, $CH_4$ | 9/2016 |
| JES | GHG60 | Jessup, MD | 39.1723 | -76.7765 | 67 | 91/49 | $CO_2$, $CH_4$ | 5/2016 |
| DER | GHG63 | Derwood, MD | 39.1347 | -77.1419 | 125 | 54/30 | $CO_2$, $CH_4$ | 5/2018 |
| CPH | GHG66 | Capitol Heights, MD | 38.8707 | -76.8537 | 50 | 44/25 | $CO_2$, $CH_4$ | 2/2018 |
| HRD | GHG67 | Herndon, VA | 38.9663 | -77.3935 | 120 | 38/27 | $CO_2$, $CH_4$ | 10/2017 |
| BWD | GHG64 | Brentwood, MD | 38.9343 | -76.9556 | 17 | 51/33 | $CO_2$, $CH_4$ | 8/2018 |
| BRK | GHG68 | Burke, VA | 38.7742 | -77.2631 | 111 | 40/24 | $CO_2$, $CH_4$ | Planned 2020 |
| | | Southeast DC | | | | | $CO_2$, $CH_4$ | |
| **Washington, D.C. and Baltimore Background Sites (5)** | | | | | | | | |
| BUC | GHG01 | Bucktown, MD | 38.4597 | -76.0430 | 3 | 75/46 | $CO_2$, $CH_4$ | 1/2015* |
| TMD | GHG61 | Thurmont, MD | 39.5768 | -77.4881 | 561 | 113/49 | $CO_2$, $CH_4$ | 5/2017 |
| SFD | GHG65 | Stafford, VA | 38.4459 | -77.5300 | 77 | 152/100/50 | $CO_2$, $CH_4$ | 7/2017 |
| BVA | GHG62 | Bluemont, VA | 39.1379 | -77.8326 | 486 | 111/50 | $CO_2$, $CH_4$ | 2/2020 |
| | | Delta, PA | | | | | $CO_2$, $CH_4$ | |

The stations in Table 1 are all situated in areas with different land use. Even among the Washington D.C. and Baltimore area urban stations, the degree of urban intensity varies, from densely urbanized areas (such as northeast Baltimore, NEB) to dense and moderately developed suburbs (such as Capitol Heights (CPH), and Derwood (DER), both suburbs of Washington, D.C.

located in Maryland). Fig. 2 indicates the intensity of development from the US Geological Survey (USGS) 2016 National
Land Cover Database (Yang et al., 2018) surrounding each urban station in the Washington DC/Baltimore network.

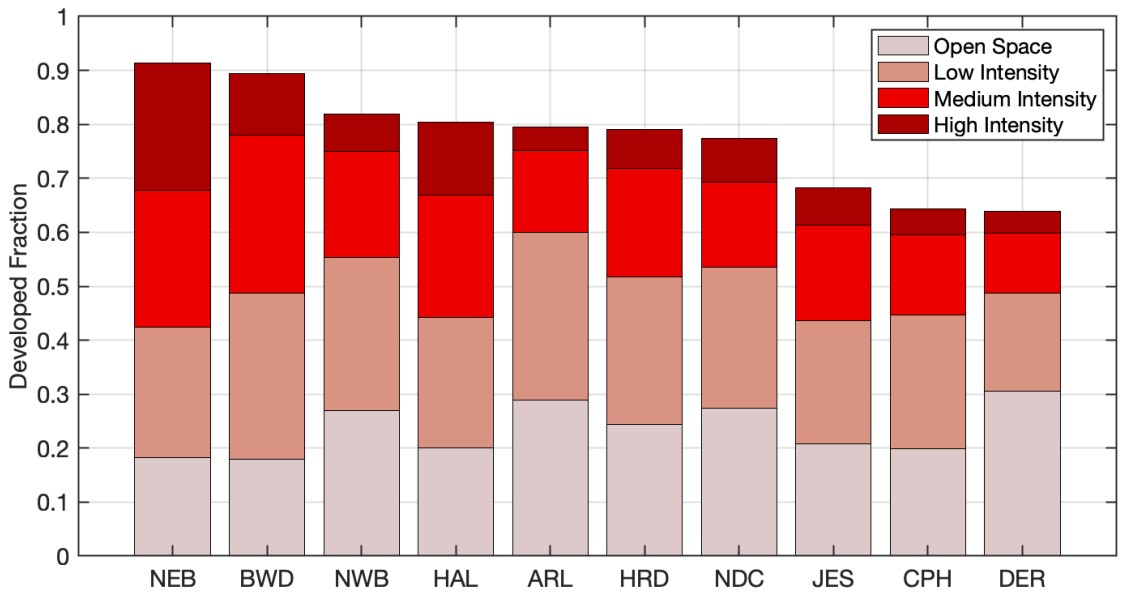

**Figure 2. Fraction of developed land cover (from the 2016 National Land Cover Database (NLCD)(Yang et al., 2018)) within 5 km**
**of each observation station in the urban regions of Washington, D.C. and Baltimore, MD.**

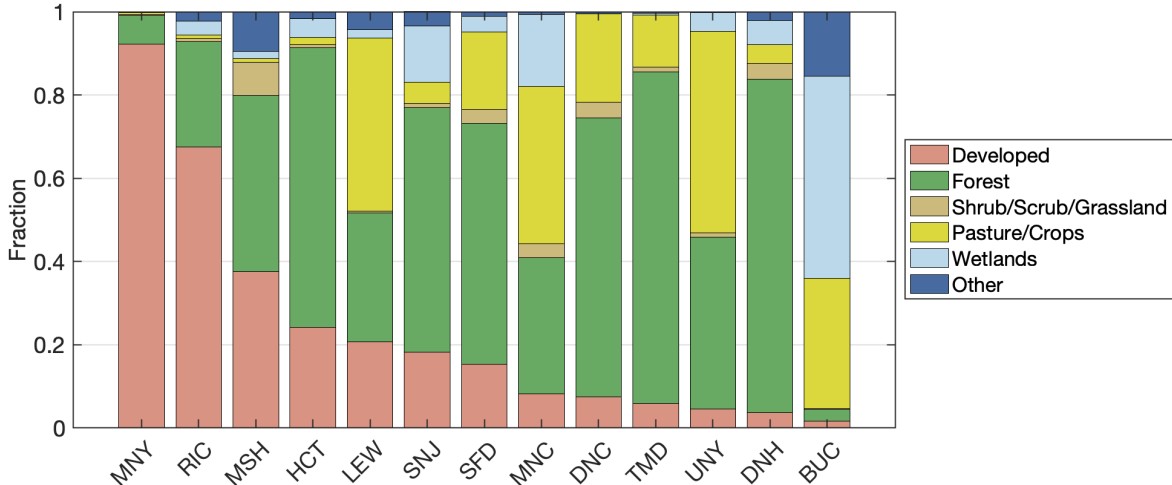

**Figure 3. Average fraction of land cover type within 5 km of regional tower sites in the Northeast Corridor network, in order of decreasing developed land. Several NLCD classifications have been grouped for clarity (e.g. Developed includes Open Space, Low, Medium, and High Intensity Developed land). SFD, TMD, and BUC are sites established to help characterize background conditions for the Washington D.C. and Baltimore urban network.**

Similar variability in land cover for the regional stations exists, as indicated in Fig. 3. The sites established to characterize background conditions for the urban network in Washington, D.C. and Baltimore (SFD, TMD, BUC) are in areas with little development: SFD and TMD are both in forested regions, while BUC is near the Chesapeake Bay and large wetland areas. The other regional sites span a range of land cover types from urban (MNY in New York City and RIC in Richmond, VA), to mostly rural and forested (DNH in Durham, NH).

## 3 Carbon dioxide, methane, and carbon monoxide measurements, instrumentation, and calibration

### 3.1 Instrumentation

The instrumentation contained in the Earth Networks (EN) system module has been described elsewhere (Welp et al., 2013; Verhulst et al., 2017); we will summarize the system here but refer the reader to those publications for further details, including additional equipment and part numbers. Figure 4 indicates the plumbing diagram of the typical tower setup. Three inlet lines reach from the sampling location on the tower into the equipment housed in a full-size rack inside a shed at the base of the tower. Typically, two inlet lines sample from the topmost level and one line samples from a lower level on the tower. Stafford, Virginia (SFD) is one exception with inlets at three different levels (50 m, 100 m, and 152 m), and a planned tower in New Jersey (Waterford Works) will have 5 inlet height levels, as indicated in Table 1. At some sites there was no space to house the equipment in existing structures, so small single- or double-rack sized enclosures were purchased and installed. Air is pulled through a filter into the inlet lines (0.953 cm (3/8") OD Synflex 1300) that are continuously flushed at ~10 L min$^{-1}$ by

aquarium pumps (Alita AL-6SA). The three air lines are connected to a rotary multi-port valve (MPV; 8-port, VICI, Valco Instruments Co. Inc.) housed within a sample control box (calibration box). Two or three calibration standards are also

connected to the MPV with 0.156 cm (1/16") OD stainless steel tubing. The control system for the MPV directs the air stream to the analyser cycling every 20 minutes through each of the three inlet lines, so that each inlet is sampled at least once in an hour, and every 22 hours through each standard (Section 3.2). The common port of the MPV is connected to a pressure controller that reduces the pressure to 80 kPa (800 mb), after which the sample (either ambient air or air from a standard gas cylinder) enters a 183 cm long Nafion dryer (Permapure, Inc., model MD-050-72S-1) where it is dried to a water vapor mole

fraction of ~0.1 % prior to flowing through the cavity ringdown spectroscopic (CRDS) analyser (Picarro, Inc., Model 2301). The lower-than-ambient inlet pressure of 80 kPa is prescribed in order to lower the flow rate of the analyser to ~70 standard $cm^3$ $min^{-1}$. At Mashpee, Massachusetts (MSH), a CRDS Picarro Model 2401 analyser is operational, and this is the only site currently also measuring carbon monoxide (CO) in addition to $CO_2$ and $CH_4$. The CRDS analysers report measurements of dry air mole fraction of each gas in air, also known as the mole fraction, i.e., moles of the trace gas per mole of dry air.

Throughout, we refer to these measurements in units of $\mu mol$ $mol^{-1}$ for $CO_2$ and $nmol$ $mol^{-1}$ for $CH_4$ and CO, following the SI recommendations (Bureau International des Poids et Mesures, 2019). Software (GCWerks, Inc.) installed on a separate mini-PC computer at each site controls the run cycle and the MPV selection valve. The data is collected on this computer and sent to the central EN data server, also running GCWerks. All data is processed on the central EN data server but additional post-processing and uncertainty assignment to hourly observations is performed at NIST. As recommended by the World

Meteorological Organization (WMO), the software has the capability of re-processing all the data from the original raw files, thus can accommodate any changes to the assigned values of the standards (due to a reference scale update, for example) at any time (WMO, 2018).

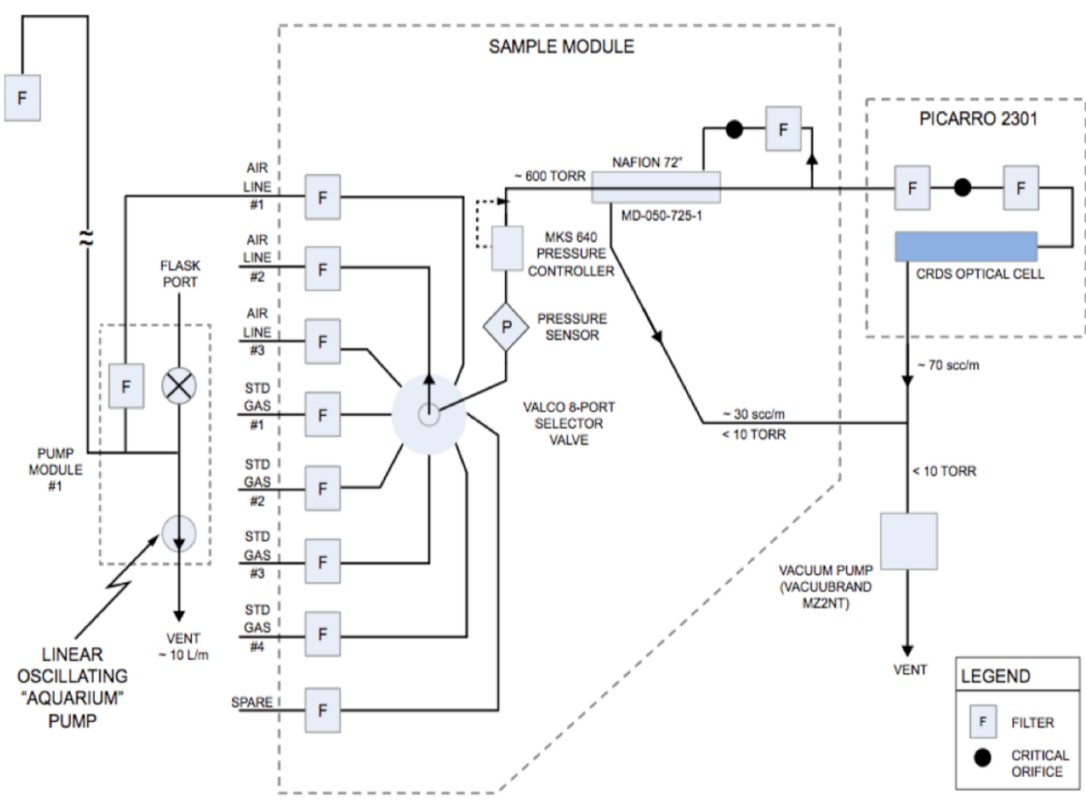

**Figure 4. Plumbing diagram for the Earth Networks sampling system implemented at the NEC tower stations. Figure replicated from Fig. S1 in Verhulst et al. (2017), adapted from Welp et al. (2013).**

### 3.2 Calibration cylinders

When the Earth Networks GHG monitoring system was established in 2011, each site hosted two calibration cylinders (standards) with ambient level dry air mole fractions as part of the original system design. This continues to be the case at most NEC sites. At the NEC sites, these standards have values close to 400 $\mu$mol mol$^{-1}$ dry air $CO_2$, 1890 nmol mol$^{-1}$ dry air $CH_4$ and 115 nmol mol$^{-1}$ dry air CO (at MSH only) and are sampled by the analyser periodically, in a sequence identical to that described for the Los Angeles Megacity network by Verhulst et al. (2017). The standards are purchased from the WMO Central Calibration Laboratory (CCL), the National Oceanographic and Atmospheric Administration's Earth System Research Laboratory (NOAA/ESRL) Global Monitoring Division in Boulder, Colorado, USA, where they have been calibrated on the WMO scales (X2007 for $CO_2$, X2004A for $CH_4$, and X2014A for CO (Zhao et al., 1997; Dlugokencky et al., 2005; Novelli et al., 2003)). One of these two cylinders serves as a standard for calibration and drift-correction while the second serves as a target tank or check standard. The target tank is used for data-quality checks and uncertainty calculations (Section 4). The residual of the target tank (the RMS difference between its value assignment when treated as an unknown and its reference value from NOAA) is a critical indicator of data quality and is monitored in order to alert the operators of any general problems

in the system such as leaks, mistakes in the assignment of MPV ports, or drift in calibration tank value. In the field, all gas standards are sampled for 20 minutes every 22 hours. In data processing, the first 10 minutes of any tank run are filtered out to allow for the system equilibration, including flushing of the regulator and tubing. In some cases when the standard runs were found not to equilibrate as quickly as desired, 15 minutes of data were filtered, until the problem could be fixed (typically either contamination or inadequate regulator flushing). The first 10 minutes of the ambient air sample following a standard run

are also filtered for equilibration, and the first one minute of each 20-minute ambient air run is filtered if it follows another ambient air run (i.e. an inlet switch). The longer flush time is desired for the standard runs because of the need to flush stagnant air remaining in the regulators and tubing when sampling from the cylinder, while the ambient air lines are continuously flushed.

At a few NEC sites (currently BWD and MSH, with more planned), a third gas cylinder is installed at the site to serve as a permanent high-concentration standard (referred to as the high standard), to improve calibration and reduce uncertainties. This standard typically contains air with a mole fraction of $CO_2$ close to 500 $\mu$mol mol$^{-1}$, $CH_4$ at approximately 2300 nmol mol$^{-1}$ to 2500 nmol mol$^{-1}$, and at MSH, CO near 320 nmol mol$^{-1}$. At MSH, this cylinder has been provided directly by NOAA/ESRL, while at BWD this cylinder was purchased as natural whole air from Scott-Marrin, Inc (now Praxair).  The Scott-Marrin air is

stripped of its original trace gases ($CO_2$, $CH_4$, CO, hydrocarbons, etc.) with $CO_2$, $CH_4$, and CO added back in to prescribed values. Several such standards have been purchased with the intent to place them at urban stations to serve as high standards after calibrating them onto the WMO scales. We note that because they are being used together with NOAA/ESRL standards in the field, it is essential that these standards also be assigned values on the same scales. This calibration is transferred in the NIST laboratory using five standards calibrated and purchased from NOAA/ESRL. The $CO_2$ in the Scott-Marrin cylinders is

isotopically different (in terms of the 12 C/13 C ratio in $CO_2$) from the ambient air tanks that are filled by NOAA/ESRL at Niwot Ridge, Colorado. However, the calibration is transferred from the NOAA standards to the Scott-Marrin gases using the same model (Picarro 2301) analyser used in the field (i.e. measuring only $^{12}CO_2$) in the NIST laboratory, effectively cancelling out the error that would be caused by this isotopic mismatch (Chen et al., 2010; Santoni et al., 2014). Thus, the $CO_2$ values assigned by NIST to these standards are effectively the total dry air mole fraction of $CO_2$ the cylinders would contain if they

were isotopically similar to the NOAA cylinders.

    Additional sites in the network also benefit from the improved two-point calibration method in cases where measurements of a high standard were performed prior to analyser deployment (NWB, NEB, JES, TMD, CPH, and HRD).  Prior to system installation at these sites, tests were conducted at the EN laboratory in which the designated analyser was set up measuring the

calibration standard, target standard, and a high-value standard at ~490 $\mu$mol mol$^{-1}$ $CO_2$, ~2560 nmol mol$^{-1}$ $CH_4$ daily for several days (enough for 3-5 measurements of 20 minutes each). This single high standard cylinder was also calibrated by and purchased from NOAA/ESRL, with assigned values on the WMO scales. These laboratory tests allow the determination of the secondary correction to the instrument response, or sensitivity, as described in Section 3.4.

The high standard gas measurements are used to perform a secondary correction (referred to as a two-point calibration) (Section 3.4) to the original one-point calibration described by Verhulst et al. (2017) and in Section 3.3., reducing the uncertainty of the measurements. We note that while in principle a secondary correction is desirable, and the uncertainty is indeed reduced by its implementation (see Section 4.2), it remains quite small relative to the signals of interest in an urban network. Deployment of high standards at all sites has not yet occurred due to both costs and logistical and operational constraints; for example, at many sites the space available for the equipment is limited and prohibits the installation of a permanent third tank. Thus, we plan to implement a round-robin procedure circulating additional standards at various values through the network to evaluate the calibrations and implement the secondary correction throughout the network. Although the current state of having two different calibration methods co-existing in the network is not ideal, we aim to implement the secondary correction throughout the network as soon as possible.

### 3.3 Drift correction and single-point calibration

Here we describe the calibration and drift correction applied to all the mole fraction data. This single-point calibration uses only a single reference value, that of the calibration standard, to correct the raw mole fractions for each gas. The equations are identical (with a few nomenclature differences) to those found in Verhulst et al. (2017). In the following analysis, $X'$ denotes a raw dry mole fraction measurement (i.e. a reported value from the CRDS analyser after internal water vapor correction), while $X$ denotes a mole fraction after some correction has been applied (drift and/or calibration, as described in the equations below). A subscript *cal* indicates the main calibration standard (usually a single ambient level standard tank calibrated by NOAA/ESRL), subscript *std* indicates any other standard tank, *tgt* indicates a standard tank that is being used as a target, and the subscript *air* indicates the sample measurement. Note that within the GCWerks software, the meanings of the abbreviations *cal* and *std* are reversed from what is defined here; we choose to use the nomenclature by Verhulst et al. (2017) here for consistency in the literature. We note that we have changed some nomenclature slightly from Verhulst et al. (2017) for additional clarity and conciseness. We refer to the drift-corrected mole fraction as $X_{DC}$, which is noted as $X_{corr}$ by Verhulst et al. (2017); we refer to the mole fraction after a secondary correction is applied as $X_{SC}$. We also refer to the assigned mole fraction of a standard by the calibration laboratory as $C$ rather than $X_{assign}$. We define the sensitivity $S$ to be the response of the analyser, or the ratio of the measured to the true value. In the case of the calibration tank, this is the ratio of the raw measured value, $X'_{cal}$, to the assigned value of the standard by the calibration laboratory on the WMO scale for the given species, $C_{cal}$:

$$S = \frac{X'_{cal}}{C_{cal}}. \tag{1}$$

When only a single calibration standard is present (which is the case at most sites in the NEC network), this sensitivity is
assumed to be constant across mole fractions, but time-varying. The sensitivity for the calibration tank is thus interpolated in
time and applied as a correction for the dry air mole fractions of $CO_2$ and $CH_4$ reported by the CRDS analyser ($X'_{air}$):

$$X_{DC,air} = \frac{X'_{air}}{S}, \tag{2}$$

where $X_{DC,air}$ is the drift-corrected air data. An alternative drift-correction is to use an additive offset, which is also interpolated
in time, rather than a sensitivity for drift correction:

$$X_{DC,air} = X'_{air} + (C_{cal} - X'_{cal}). \tag{3}$$

Measurements from MSH that include a high value cylinder suggest that the single tank drift correction performs (very slightly)
better using the ratio correction (Eq. 2) than the difference method (Eq. 3) for $CO_2$ and $CH_4$, while the opposite is true for CO
(Fig. 5), so the difference method is used only for CO in our network.

The calibration standard mole fractions are interpolated in time between subsequent runs in order to apply the above corrections
to the air data, thus removing drift in the instrument's response. This drift-corrected fraction is reported in the hourly data files
for sites and time periods where no range of concentrations is available in the standard tanks.

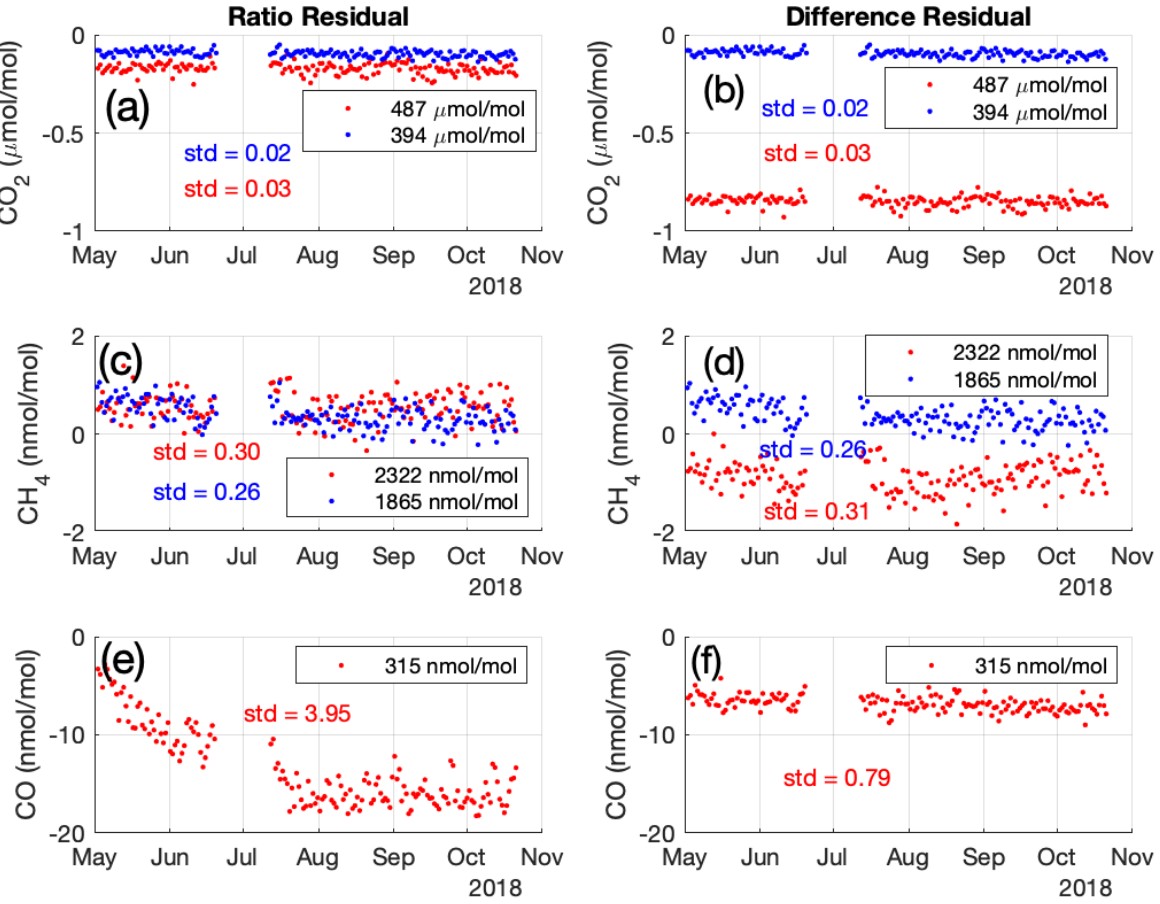

Figure 5. Time series of standard tank run residuals (i.e. $X_{DC} - C$) for $CO_2$ (a,b), $CH_4$ (c,d) and CO (e,f). $X_{DC}$ is calculated using a single calibration tank (not shown) and the ratio method (Eq. 2) on the left (a,c,e) and the difference method (Eq. 3) on the right (b,d,f). Assigned tank values are shown in the legend; one tank was not calibrated for CO so only the residuals of the high concentration tank at 315 nmol mol$^{-1}$ are shown. The residual magnitude is smaller for $CO_2$ and $CH_4$ using the ratio method, but the standard deviations (variability) are similar using both methods. For CO, both the magnitude of the residual and also the standard deviation are smaller using the difference equation; the ratio equation does not properly account for the drift in the analyser at the start of the time series (May-June). Data shown are from MSH; a measurement gap exists in July.

### 3.4 Multiple-point calibration

At some sites and for some time periods, a higher-mole-fraction standard is available, and a second-order correction can be made to the instrument sensitivity, accounting for the sensitivity being a function of mole fraction. Usually in the field, this correction employs only one additional standard, the higher-mole-fraction standard, so that it is a two-point calibration; here we describe the general procedure for applying a correction using multiple standards at a range of concentrations. This is applied as a second-order correction to the drift-corrected air data. In general, if a range of standard concentrations is available,

the correction in GCWerks is applied as described below. First, a drift-corrected sensitivity ($S_{DC}$) is calculated for each standard when it is measured, which is the ratio of the drift-corrected mole fraction of that standard ($X_{DC,std}$, based on Eq. (2) for $CO_2$ and $CH_4$ or Eq. (3) for CO) to its assigned value:

$$S_{DC,std} = \frac{X_{DC,std}}{C_{std}}. \tag{4}$$

For the calibration standard, this value is necessarily equal to 1, but measurements of standard tanks with different assigned values indicate that the instrument sensitivity is dependent on the composition of the sample gas (in this case, the mole fraction of the standard tank). In laboratory calibrations, we find that the drift-corrected sensitivity defined in Eq. (4) is a linear function of the mole fraction ratio to the calibration gas ($X'/X'_{cal}$), so we use a linear fit to the range of standards to determine the slope $m$ and intercept $b$:

$$S_{DC} = m\left(\frac{X'}{X'_{cal}}\right) + b. \tag{5}$$

In this fit, we force $m + b = 1$ by fitting a slope $m$ and then setting $b = 1-m$ in order to maintain the proper relationship for the calibration tank itself, when $S_{DC,cal} = 1$. Applying this fit to the air data, the final air mole fraction $X_{SC,air}$ is determined from:

$$X_{SC,air} = \left(\frac{X_{DC,air}}{S_{DC}}\right). \tag{6}$$

In the NEC tower network, there are no sites with multiple standard tanks at various concentrations. At several sites, there are measurements of a single high-concentration standard (*hstd*) in addition to the calibration and target standards. The high standard measurements are either performed in the laboratory before the instrument is deployed to the field, or in the field if the third standard is permanently installed (Section 3.2). The above secondary correction is applied using only two tanks to perform the fit and obtain the drift-corrected sensitivity. In this special case, the fit has zero degrees of freedom with no residuals. The correction parameters (slope and intercept) are determined based on measurements over time or single measurements in the laboratory prior to a specific analyser deployment. The correction is applied to the data from the site for a time period that is specified, i.e. it is not automatically applied based on daily measurements of the high standard. It is determined by the science team and applied for the time period that is appropriate. This is necessary to avoid applying the wrong correction if an analyser is replaced or if there are changes made to the analyser that might affect its calibration response. At eight sites where a high standard has been measured at any point (MSH, BWD, NWB, NEB, JES, TMD, CPH, and HRD), slopes and intercepts have been determined and the correction has been applied to the data. At stations with no high standard measurements we rely on the single tank drift-correction described in Section 3.3.

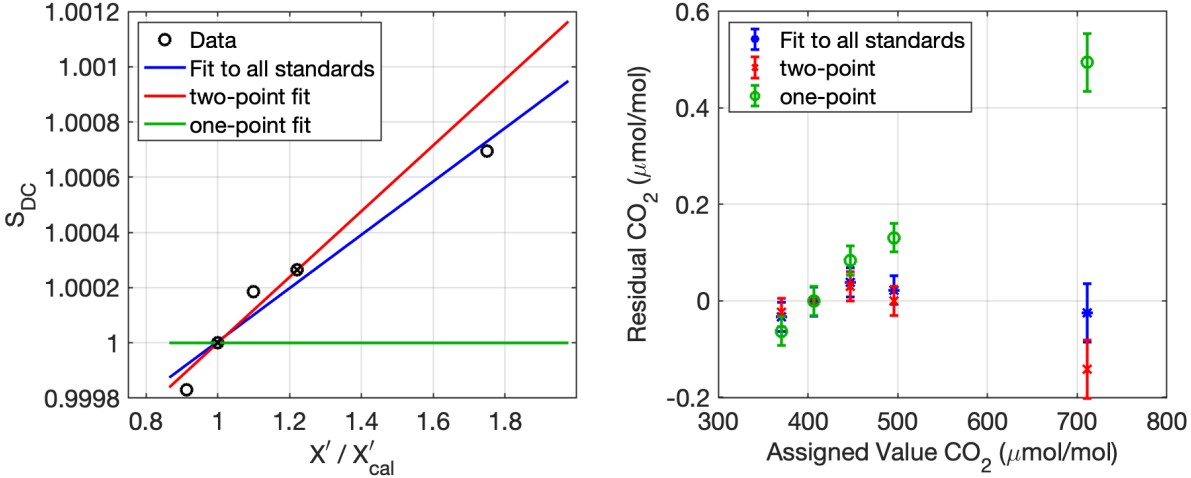


**Figure 6. Example of a laboratory calibration of a CRDS analyser with five standards of different assigned $CO_2$ mole fractions. (a) Secondary correction of drift-corrected sensitivity using either two (red) or all five (blue) standards. Green line at 1 indicates the assumed sensitivity when only a single standard is used. (b) Residual of each type of fit; error bars represent 1-sigma reproducibility stated by NOAA/ESRL. The simple single-tank drift correction results in the green circles as residuals; these residuals were used in**
**the Verhulst et al. (2017) analysis to estimate the extrapolation uncertainty of the single-point correction. Red x symbols are the residuals of a fit to two standards, and blue asterisks the residuals of the fit to all five standards.**

Laboratory tests with multiple standards with the same model instrument used in the network (Picarro 2301) were performed to assess the relative improvement of a fit to two standards over a fit to a single standard. Figure 6(a) illustrates the fit of the
drift-corrected sensitivity ($S_{DC}$) to two standards (red line) vs. all five standards (blue line) for $CO_2$, along with corresponding residuals in (b). As was shown by Verhulst et al. (2017) for multiple analysers, the fit to a single standard has a linearly varying residual that is typically 0.1 to 0.2 μmol mol$^{-1}$ at 100 μmol mol$^{-1}$ above the calibration standard value (green circles, (b)). The average slope of the one-point residual from multiple tests is used by Verhulst et al. (2017) to estimate the uncertainty of the single-point calibrations (called the extrapolation uncertainty, $U_{extrap}$), described in Section 4.1. Performing the additional
correction using a high standard shows improvement in the residuals of the fit (Fig. 6(b)), while using all five standards only improves the residuals incrementally. The two-point correction (red) in this figure was applied using the 406 μmol mol$^{-1}$ tank as the calibration and the 496 μmol mol$^{-1}$ tank as the high standard; thus, the measurement at ~711 μmol mol$^{-1}$ is an extrapolation of the two-point fit. The residuals at values between the calibration and high standard are very small, equal to or below the uncertainty (reproducibility) of the scale reported by NOAA; this was confirmed for other analysers and other
species.

The improvement in calibration from the secondary correction is quite small compared to the signals and gradients of interest in our network. For example, when considering the enhancement between the rural site TMD and a polluted urban site, HRD, the calibration method makes a median difference of 0.4% for $CO_2$ and 0.3% for $CH_4$ (over all hours over one calendar year).

We intend to implement this calibration throughout the network through deployment of additional standards and periodic traveling calibrations when permanent installation is not practical for logistical reasons.

### 3.5 Data quality and processing

Automated data filtering is performed within the GCWerks software with parameters set identically to those extensively described by Verhulst et al. (2017) for the Los Angeles Megacities network. For example, individual measurements that are
outside limits for cavity temperature, cavity pressure, and during transitions between sample streams are filtered. The data is automatically downloaded from each site's Linux PC to the central EN Linux server, where it is processed automatically every hour. We note that all mole fraction assignments can be re-calculated by the GCWerks software from the archived raw files if required due to a change in filtering or flagging, or in assignment of a standard tank, for example, in the case of a scale change by the CCL. The data files exported from GCWerks contain 1-minute, 5-minute, and 20-minute averaged air data, as well as
separate files with 1 min, 5 min and 20 min averages of all standard runs. Individual or groups of 1-minute data points are flagged manually by EN or NIST researchers in the GCWerks if there is cause (e.g. a site visit that disrupted the sample stream, or a leak in the line, etc.). Some additional quality checking is performed at this stage, specifically checking for systematic differences between measurements from two different inlets at the same height and checking for inconsistency in the difference between measurements at different heights. For example, if the lower inlet is systematically reading lower $CO_2$ than the upper
inlet, especially at night, it would indicate that the inlet lines may be switched (mislabelled) or there is a leak occurring. These indications would be then verified by a field technician and the data either re-processed or flagged accordingly. Filtered and flagged points are excluded from the subsequent averaging exported by GCWerks. The 1-minute air data files and 20-minute standards data files are post-processed at NIST to calculate hourly averages from each air inlet level, and to assign uncertainties to each hourly average (Section 4). Data from the two top-level inlets, when at the same height, are combined for inclusion
into the hourly average. Thus, because of the 20-minute cycling through the three inlets (Section 3.1), hourly averages at the upper inlet include approximately 40 minutes of measurements, and for the lower inlet only 20 minutes (fewer if a calibration occurs). Publicly released hourly data from this second-level processing is contained in separate files for each species and each level for each site. The files contain the hourly average mole fraction (i.e. mole fraction) along with its uncertainty, standard deviation, and number of 1-minute air measurements included in that particular hourly average. These last two quantities are
provided so users can determine the standard error of the hourly means, in terms of the observed atmospheric variability within the hour. Observations at higher frequency and standard tank data are available by request.

### 3.6 Comparison with measurements of NOAA whole air samples

Ongoing whole air sampling in flasks at several of the NEC sites by NOAA Earth System Research Laboratory's Global Monitoring Division (NOAA/GMD) provides a check on the quality of the in-situ measurements. The flasks are analyzed for
$CO_2$, $CH_4$, and CO, among a suite of additional trace gases and isotopes which are not discussed here. The flask sampling equipment draws air from one of the inlet lines at the top of the tower that is also shared by the in-situ continuous measurement

equipment (as indicated by the flask port in Fig. 4). The flask measurements are otherwise independent from the continuous in-situ measurements. Flask samples at LEW and MSH are collected over a period of 10-30 seconds (Sweeney et al., 2015; Andrews et al., 2014), while flask samples integrated over one hour are collected at TMD, NEB, NWB, and BWD (Turnbull et al., 2012) specifically as part of the Northeast Corridor project. All flask samples are taken in mid-afternoon local time (usually 19 UTC). Comparisons at all the sites with available data indicate good agreement with little or no bias in the mean over the time period of the comparison, with the exception of CO at MSH, which shows a consistent bias with a median of 8 nmol mol$^{-1}$, which is larger than the 1-sigma uncertainty assigned to either measurement (described in Section 4) and the standard deviation of the offsets themselves (Table 2). Target tank residuals for CO in this period range from 1 to 7 nmol mol$^{-1}$, depending on the cylinders installed, indicating that some of this difference at least is caused by the calibration standard assigned value (possibly due to cylinder drift in time between the NOAA calibration and deployment to the site). Similar differences between NOAA flasks and in-situ CO measurements were reported in Indianapolis (Richardson et al., 2017). This result requires further investigation, by sending the cylinders for recalibration and/or deploying different standards to the station. A significant bias in the $CH_4$ offset at NWB is also apparent, at a mean of -5.5 nmol mol$^{-1}$ but a median of -1.7 nmol mol$^{-1}$, the result of a single outlier at -30 nmol mol$^{-1}$, but with only 17 samples compared. BWD did not have any samples at the time of this writing so we compare only LEW, MSH, TMD, NEB, and NWB.

Table 2 also reports the mean uncertainty, intended as a metric for comparison of the standard deviation of the offsets. For each flask sample, this uncertainty is the quadrature sum of the continuous data uncertainty (described in Section 4) at that hour, the standard deviation of the 1-minute averages in the continuous data during that hour, and the uncertainty expected in the flask measurement, estimated here as 0.04 μmol mol$^{-1}$ for $CO_2$, 1.12 nmol mol$^{-1}$ for $CH_4$, and 0.59 nmol mol$^{-1}$ for CO. The values for the flask uncertainty are from Table 1 in Sweeney et al. (2015), which reports the average offset between measurements of surface network and 12-pack flasks (such as those used for the NEC) filled with identical air after a short-term storage test. For $CO_2$, flask offsets can be larger than indicated by those dry-air laboratory tests (Sweeney et al., 2015; Andrews et al., 2014; Karion et al., 2013), but we use 0.04 μmol mol$^{-1}$ regardless, because the average uncertainty in Table 2 is dominated by the atmospheric variability term and increasing the $CO_2$ uncertainty in the flasks to 0.1 μmol mol$^{-1}$ (for example) does not change the values significantly.

Standard deviations of the offsets (Table 2) show that there is quite a bit of scatter in the results, especially at the more urban sites that exhibit a lot of variability in the continuous data. For comparison, Turnbull et al. (2015) report agreement for $CO_2$ between the same flask system and continuous in-situ measurements in Indianapolis as 0.04 μmol mol$^{-1}$ (mean) with a standard deviation of 0.38 μmol mol$^{-1}$, somewhat smaller than observed at our sites. The standard deviation of offsets is usually lower than the average uncertainty, however, with the exception of $CO_2$ at MSH and LEW, the two sites for which the flask samples are not integrated over an hour. It is likely that the large variability seen over an hour is the reason for the large scatter in the

offsets. Because the in-situ continuous measurements do not cover the entire hour of sampling (at the top level, the hourly average is typically the mean of only 40 minutes), the variability may not be captured in the mean uncertainty reported here, and has a larger impact on the comparison than it would if the continuous hourly average was based on the full hour of observations. For example, a large plume or spike in concentration during a given hour might occur while the continuous system is sampling from the lower inlet, and thus would not be included in the hourly average from the continuous system, while it would be included in the full one-hour flask sample.

**Table 2. Offsets (in-situ - flask) between continuous in-situ and NOAA/GMD flask measurements. $CO_2$ offsets are reported in $\mu$mol mol$^{-1}$, $CH_4$ and CO in nmol mol$^{-1}$. Continuous in-situ CO is only measured at MSH. The average uncertainty column for each gas indicates the 1-sigma uncertainty (summed in quadrature over flask uncertainty, in-situ uncertainty, and atmospheric variability over the hour) averaged over the flask samples. See text for discussion.**

| Site | Number of flask samples | $CO_2$ mean offset | $CO_2$ median offset | $CO_2$ standard deviation of offsets | $CO_2$ mean unc. | $CH_4$ mean offset | $CH_4$ median offset | $CH_4$ standard deviation of offsets | $CH_4$ mean unc. | CO mean offset | CO median offset | CO standard deviation of offsets | CO mean unc. |
|------|------|------|------|------|------|------|------|------|------|------|------|------|------|
| MSH | 163 | 0.02 | -0.02 | 0.65 | 0.43 | -0.2 | -0.2 | 2.3 | 2.1 | -9.0 | -8.2 | 5.9 | 6.0 |
| LEW | 315 | 0.01 | -0.07 | 0.88 | 0.68 | 1.1 | 0.4 | 8.4 | 7.8 | -- | -- | -- | -- |
| TMD | 80 | 0.17 | 0.15 | 0.51 | 0.69 | 0.0 | 0.5 | 5.5 | 5.2 | -- | -- | -- | -- |
| NEB | 32 | -0.09 | 0.08 | 0.75 | 1.02 | -0.5 | 0.6 | 6.8 | 13.4 | -- | -- | -- | -- |
| NWB | 17 | -0.01 | 0.02 | 0.73 | 0.99 | -5.5 | -1.7 | 9.8 | 8.3 | -- | -- | -- | -- |

## 4 Uncertainty

The data set includes an uncertainty estimate on each hourly average data point, consistent with recommendations from the WMO (WMO, 2018). This uncertainty is our estimate of the uncertainty of the measurement itself and does not include atmospheric variability or assess the representativeness of the measurement of a true hourly mean.

### 4.1 Uncertainty of hourly mole fraction data

Verhulst et al. (2017) outlined a method for calculating an uncertainty on mole fraction measurements when using the single tank calibration correction (drift correction). Here we present a brief overview but refer the reader to that paper for further details. All uncertainties are standard uncertainties, i.e. 1-sigma or $k$=1. In the analysis below, we assume independent uncorrelated error components, given no evidence to the contrary and no physical reason to believe that they should be correlated; therefore we sum the various components of the uncertainty in quadrature.

The uncertainty on the final mole fractions ($U_{air}$) is expressed as the quadrature sum of several uncertainty components:

$$(U_{air})^2 = \left(U_{extrap}\right)^2 + (U_{H2O})^2 + (U_M)^2 \tag{7}$$

where $U_{H2O}$ is the uncertainty due to the water vapor correction, $U_M$ is a measurement uncertainty, and $U_{extrap}$ is the uncertainty of the calibration fit when assigning values relative to a single standard tank (more detail on this can be found later in this section and in the following section). $U_M$ encompasses errors due to drifting standard tank measurements ($U_b$), short-term precision ($U_p$), and error in the calibration standard's mole fraction assignment by the calibration laboratory ($U_{scale}$):


$$(U_M)^2 = \left(U_p\right)^2 + (U_b)^2 + (U_{scale})^2. \tag{8}$$

Here we note that $U_p$ for $CO_2$ and $CH_4$ is assigned as described by Verhulst et al. (2017), as the standard deviation of the individual measurements during each 1-minute average during a calibration, but for CO it is assigned as the standard error

(standard deviation divided by the square root of the number of samples in the mean), based on Allan variance tests (not shown) indicating that the precision of the CO measurement increases with the number of points used in the average. If no calibrations have been performed over an entire calendar year, $U_p$ is set to the $10^{th}$ percentile of the standard deviation of air measurements, and $U_b$ is set to a default value of 0.1 µmol mol$^{-1}$, 0.5 nmol mol$^{-1}$, and 4 nmol mol$^{-1}$ for $CO_2$, $CH_4$, and CO respectively. This default value for $U_b$ is based on an upper limit of values that are observed in the network; typically, $U_b$ is much smaller than

these values (Verhulst et al., 2017). In the current data set, this has only occurred once: there were no calibrations run at MNC over the entire 2015 calendar year, but we have no knowledge of abnormal operations or changes during this period, with analyser sensitivity being similar before and after this period.

Because these uncertainty components are also tested through the use of a target tank, or check standard, the uncertainty $U_M$

is assigned as the root-mean-squared of the target tank errors when those exceed the sum of the uncertainties above:

$$U_M = U_{TGT} = \sqrt{\frac{\Sigma\left(\left(x_{DC,TGT} - c_{TGT}\right)^2\right)}{N}}. \tag{9}$$

This residual is calculated by GCWerks, and the root mean square residual is interpolated in time as a moving 10-day average.

If a target tank has not been run through the system for 10 days or longer, $U_{TGT}$ is set to a default value that is currently set to 0.2 µmol mol$^{-1}$, 1 nmol mol$^{-1}$, and 6 nmol mol$^{-1}$ for $CO_2$, $CH_4$, and CO, respectively, based on typical maximum values for this uncertainty calculated from many sites over several years. The target tank in the field generally has a concentration value very similar to the calibration tank, so this residual is a good estimate of the uncertainty caused by the precision, baseline changes, and tank value assignment. However, it is not a good indicator of uncertainty at mole fractions different from that of the

calibration tank. Therefore, we assign an added uncertainty component, $U_{extrap}$, indicating the uncertainty that increases as the measurement value moves farther from the value of the calibration tank in the case of a single calibration standard. This was

found to be a linear relationship for a series of similar model analysers that were tested in a laboratory, and the uncertainty was described as:

$$U_{extrap} = |\varepsilon| |X_{DC,air} - C_{cal}|$$ (10)

See Verhulst et al. (2017) for details on determining the unitless slope of the uncertainty, epsilon ($\varepsilon$), which is currently assigned as 0.0025, 0.0031, and 0.0164 for $CO_2$, $CH_4$, and CO, respectively, for all data that is only drift corrected (i.e. not using a high standard).

## 470 4.2 Uncertainty for observations with additional standards available

When a high standard tank is available and the secondary correction described in Section 3.4 is applied, the uncertainty analysis remains similar, but the uncertainty $U_{extrap}$ from Eq. (7) and Eq. (10) is replaced by an uncertainty in the two-point fit, $U_{fit}$. To estimate this uncertainty for $CO_2$ and $CH_4$, we use the reported uncertainty of the assigned value of the high standard and calibration standard tanks, $U_{scale}$, (typically 0.03 µmol mol$^{-1}$ $CO_2$ and 0.5 nmol mol$^{-1}$ $CH_4$ at 1-sigma) along with an estimate 475 of the precision of the analyser, $U_p$, to estimate an uncertainty on the drift-corrected sensitivity of the high standard, $U_{SDC,hstd}$, using standard propagation of errors (black error bar, Fig. 7(a)). We note that in the case where the value assigned to the high standard is through a propagation of the WMO scale at NIST, the assigned value has additional uncertainty; i.e. $U_{scale}$ includes both the uncertainty that NOAA assigned to the cylinders used for the assignment and the uncertainty from the laboratory fit at NIST. This second uncertainty is equal to the standard deviation of the residuals of the fit and it is added in quadrature to 480 the NOAA uncertainty.

We note that the analysis described below assumes uncorrelated independent errors. We express the slope of drift corrected sensitivity ($m$) and the overall drift-corrected sensitivity ($S_{DC}$) as functions only of the drift-corrected sensitivity of the high standard, $S_{DC,hstd}$:


$$m = \frac{S_{DC,hstd} - 1}{X'_{hstd}/X'_{cal} - 1}$$ (11)

$$S_{DC} = m \left( \frac{X'}{X'_{cal}} - 1 \right) + 1.$$ (12)

This second equation uses $b = 1 - m$. Here we do not include uncertainty in the x-coordinate, i.e. $X'/X'_{cal}$. Uncertainty in the slope 490 is thus:

$$U_m = \left| \left( \frac{U_{SDC,hstd}}{\left( {X'}_{hstd}/{X'}_{cal} \right) - 1} \right) \right|. \tag{13}$$

We propagate the uncertainty in the drift-corrected sensitivity of the high standard, $U_{SDC,hstd}$, to the overall drift corrected sensitivity of all the air values using Eq. (14), and then to the two-point corrected air data by propagating through to obtain Eq. (15).

$$U_{SDC} = U_m \left( {X'}/{X'}_{cal} - 1 \right)$$

$$= \left| \left( \frac{U_{SDC,hstd}}{\left( {X'}_{hstd}/{X'}_{cal} \right) - 1} \right) \left( {X'}/{X'}_{cal} - 1 \right) \right| \tag{14}$$

$$U_{XSC,air} = U_{fit} = \left| \frac{U_{SDC}}{S_{DC}} \right| X_{SC,air}. \tag{15}$$

To evaluate the use of standard propagation of errors, we also use a bootstrap to estimate the uncertainty using the laboratory calibration shown in Fig. 6 by randomly selecting two tanks of the five tanks from the test to calculate 1000 versions of the correction (blue shading shows the standard deviation of the result, Fig. 7). For this test, the calculated 1-sigma uncertainty (red shading) was similar to the 1-sigma bootstrap uncertainty (slightly larger for $CO_2$ and slightly smaller for $CH_4$ (not shown)). This comparison indicates that the estimated uncertainty using the equations above compares reasonably well with the uncertainty we would derive from a bootstrap analysis, which gives us confidence in our methodology.

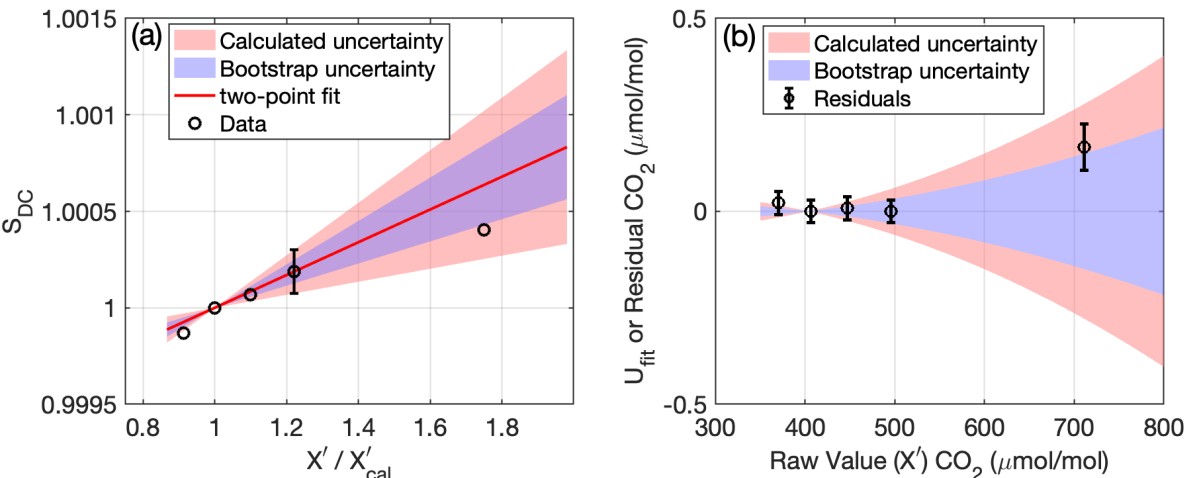

**Figure 7. Uncertainty (1-sigma) in fit for two-point calibrations. (a) two-point fit to drift-corrected sensitivity ($S_{DC}$) (red line) with uncertainty (red shading) calculated using the uncertainty in the high standard (black circle with error bar). Blue shading shows**

**uncertainty calculated using a bootstrap conducted by randomly selecting sets of two standards from the laboratory test (black circles) to calculate the slope. There is no uncertainty at 1 because the drift-corrected sensitivity is defined as equal to one at the value of the calibration standard. (b) Uncertainty in final $CO_2$ as a function of raw $CO_2$; red and blue shading as in (a).**

The uncertainty in $S_{DC}$ leads to the estimate of the fit uncertainty, $U_{fit}$, shown in Fig. 7(b). To implement this uncertainty across all times and towers, we calculate it assuming a fixed nominal value of the high calibration standard of 490.50 µmol mol$^{-1}$ $CO_2$ and 2560.61 nmol mol$^{-1}$ $CH_4$. This is based on the value of the high standard that was in residence in the Earth Networks laboratory when several of the CRDS analysers were tested and assigned two-point calibration corrections. We use the site-specific (instrument-specific and period-specific) slope and intercept that are applied to the data (which are static over the time period they are applied), and the value of the calibration tank, to calculate the remainder of the values required for the uncertainty analysis.

Only one site so far, MSH, measures continuous CO, and the history of standard tanks there indicates significant uncertainty in tank value assignments with high target tank residuals and corresponding $U_{TGT}$ relative to errors in slope. We have chosen not to implement the two-point calibration at this site for CO, because the range of slopes of $S_{DC}$ includes one, i.e. the correction is so small that the uncertainty dwarfs the correction.

Mean absolute residuals of the two-point fit for nine laboratory calibrations analysed (7 tested at NOAA/ESRL and described by Verhulst et al. (2017) Table S2 and two additional units at NIST) average to 0.03 µmol mol$^{-1}$ for $CO_2$ between the calibration and high standard, and larger for the test that included an even higher-concentration tank, shown in Fig. 7 at ~711 µmol mol$^{-1}$ for $CO_2$. The fit uncertainty encompasses (at 1-sigma) this residual as well (Fig. 7(b)). The residuals at lower values can be explained by the uncertainty in the measurement (precision) and uncertainty in value assignment of the tanks. For CO, only eight tests were available, with a mean residual inside the range of the calibrations of 1.1 nmol mol$^{-1}$, higher than the reported reproducibility from NOAA of 0.4 nmol mol$^{-1}$ (all values are noted here at 1-sigma although they are given by NOAA at 2-sigma). This larger residual is likely caused by the lower precision of the analysers for CO but also could be caused by larger uncertainty in the tank assignments, possibly due to drift in the mole fraction of the tanks themselves. We intend to conduct additional tests outside the two-point calibration range with additional analysers and tanks to evaluate and possibly update this uncertainty component, $U_{fit}$, as needed, and especially focus on CO if/when additional CO measurements are added to our network.

## 5 Network observations

Here we show some observations and time series of $CO_2$ and $CH_4$ from the NEC in-situ tower network, focusing on data coverage, vertical gradients, and observed differences between urban and rural or outer suburban sites.

## 5.1 Data coverage and network expansion

The NEC network is continuously growing, with sites coming online at different times. Figure 8 shows the availability of hourly observations as the various sites have come online.

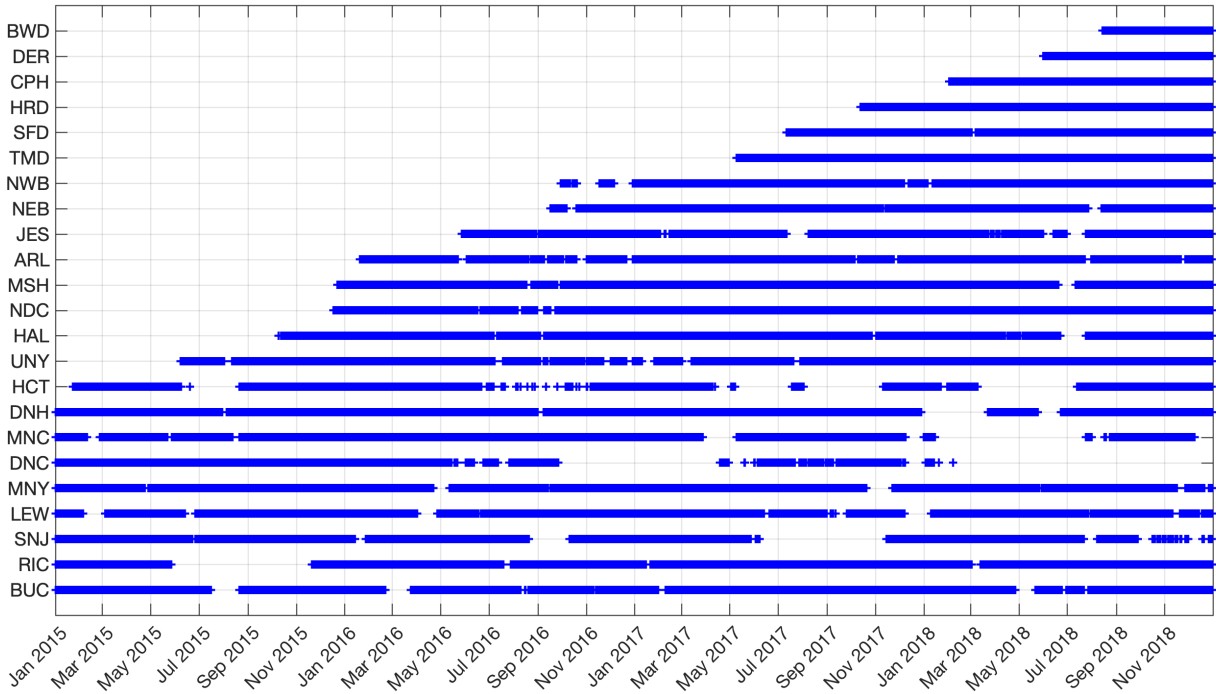

**Figure 8. Data (CO₂ and CH₄, and CO for MSH only) availability from the various NIST-EN tower sites in the Northeast Corridor network included in this data release. Gaps represent data outages due to various failures (analyser, communications, etc.).**

## 5.2 Vertical gradients

Observations in global trace gas measurement networks (e.g. AGAGE, GGRN) are specifically sited far from local sources or strong sinks, to ensure that air reaching the site is representative of the large spatial scales of interest to a global study. This allows the observations to be more easily interpreted by a coarser global model (e.g. Peters et al. (2007)). In urban networks, it is desirable to measure trace gas concentrations closer to sources so that finer spatial gradients can be used to inform emissions estimates at urban scales. However, a balance must be struck between the necessity to observe and distinguish sources that are in close proximity to each other and the ability of a transport and dispersion model to simulate the observations. In some instances, novel ways to simulate observations at low heights above ground level and in very dense networks have been used to resolve this problem (Berchet et al., 2017). In the NEC urban network in Washington DC and Baltimore, the tower sites were selected to be between 50 m and 100 m above the ground given the desire to place a tower in a specific location (as identified in an initial network design study by Lopez-Coto et al. (2017)). Inlets at two (or three, at SFD) heights on the tower give some insight as to the proximity of each tower to sources whose emissions are not always vertically well-

mixed by the time they reach the inlets, depending on atmospheric stability conditions. Here we report average vertical gradients, determined using the observations at different levels, for the urban and background sites in our network. These

565 gradients were calculated using hourly average data from each level, but because the instruments are only sampling from one level at any given time and cycling between them, there is an assumption of measurements averaged in a given hour to be representative of the entire hour. Because different towers have different inlet heights and different vertical spacing between the lower and upper inlet, here we compare three urban sites (ARL, NDC, and JES) with inlets at similar heights, ~90 m and ~50 m agl. We define the gradient as the mole fraction of $CO_2$ or $CH_4$ at the topmost inlet minus that of the lowermost inlet

divided by the distance between them, so that a negative gradient indicates a higher concentration at the lower inlet (the most common case).

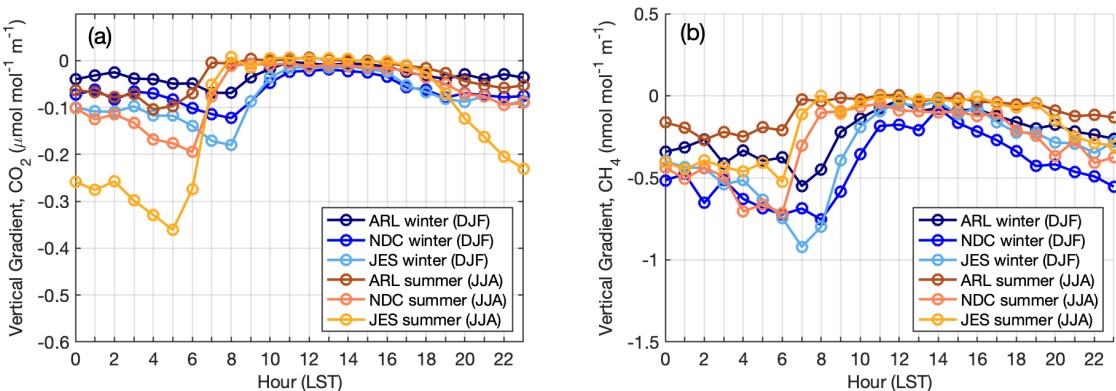

**Figure 9. Diurnal cycle of vertical gradients in $CO_2$ (a) and $CH_4$ (b) for urban towers in the Washington/Baltimore area, averaged**
**over 2015-2017 in winter (blue) and summer (orange), with shading indicating 1-sigma standard deviation among sites. Some of the spread can be caused by sampling in different years at the different sites. Sites included are: HAL, ARL, NDC, NEB, NWB, and JES. HRD was excluded due to lack of data in this period.**

Analysis of the diurnal cycle of the vertical gradient at urban sites in the Washington / Baltimore area (Fig. 9) indicates different characteristics in summer vs. winter. These differences are most likely caused by different meteorology and possible seasonal

differences in timing of fluxes, especially for sites influenced by the urban biosphere. Greater turbulent mixing in summertime boundary layers and different timing in the boundary layer growth and collapse mostly dominate the seasonal differences. This analysis shows that at these three sites the wintertime average gradient in mid-afternoon hours (defined based on these figures as 11-16 LST) is approximately -0.016 $\mu$mol mol$^{-1}$ m$^{-1}$ for $CO_2$ (-0.105 nmol mol$^{-1}$ m$^{-1}$ for $CH_4$), which translates to a -0.8 $\mu$mol mol$^{-1}$ (-5.2 nmol mol$^{-1}$ for $CH_4$) difference between levels spaced 50 m apart; this is not an insignificant gradient. At

other urban sites with shorter towers, they can be even larger. These observations can help evaluate vertical mixing in transport and dispersion models that might be used to estimate emissions, or to identify times when modelled and observed vertical gradients agree. Large vertical gradients overnight into the early morning at all sites and seasons are indicative of local sources (likely mostly anthropogenic but also including respiration from the biosphere) influencing the observations at these times

when there is stable stratification in the boundary layer and concentrations are higher near the surface. The larger $CO_2$ gradients overnight in summer compared to winter periods suggest a strong respiration signal at these urban sites, with a large degree of variability between sites indicated by large spread. Night-time $CH_4$ gradients are slightly larger in winter than summer, possibly reflecting greater wintertime anthropogenic $CH_4$ emissions, or possibly due to seasonality in mixing layer heights.

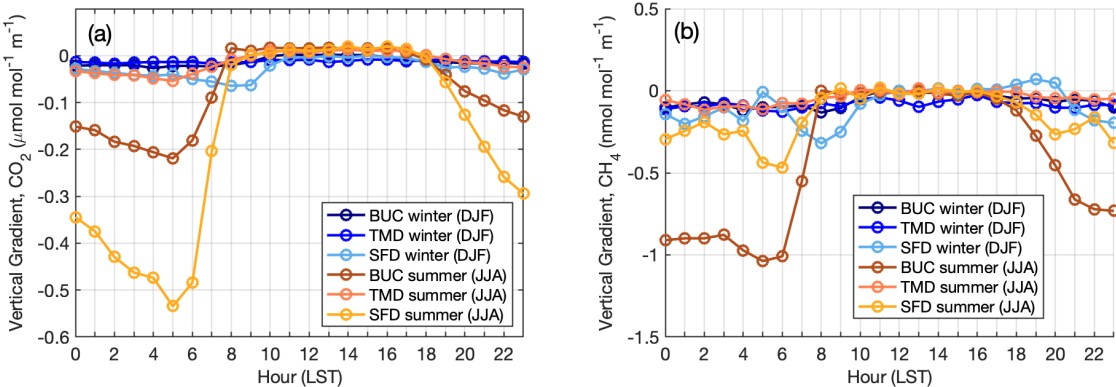

**Figure 10. Diurnal cycle of vertical gradients in $CO_2$ (a) and $CH_4$ (b) at the three background towers for the Washington / Baltimore region, in summer (orange shades) and winter (blue shades).**

The diurnal cycle of the vertical gradients from the sites identified as background stations for the Washington/Baltimore urban network shows large variability in summertime gradients between the three stations (Fig. 10). Stafford, VA (SFD) shows that the surrounding biosphere causes relatively large gradients in night-time and early morning hours at this low-density suburban site. These are apparent at Bucktown, MD (BUC) as well, but much less so at Thurmont, MD (TMD), a forested site in western Maryland. The large difference between summertime early morning vertical $CO_2$ gradients at SFD and TMD, despite the similar surrounding land use (mostly deciduous forest, Fig. 3), might be caused by the elevation difference, as SFD is close to sea level while TMD is on a ridge at 561 m elevation. BUC observations show larger $CH_4$ gradients in summer, due to surrounding wetlands and agriculture (Fig. 3). Wintertime gradients are near zero at all hours at all three of these sites, indicating that they are far from local anthropogenic sources of either gas. We note that the top inlet height at BUC is lower, at 75 m, than at SFD or TMD (100 m and 111 m), while the lower inlet is similar for all three (~50 m). For SFD (inlets at 152, 100, and 50 m), we use the 100 m and 50 m inlets to define the gradient, to be more consistent with the inlet heights of the other towers (Table 1).

## 5.3 Urban and rural differences in seasonal cycles

Here we continue to describe the network in terms of differences between rural (background) and urban stations, determining typical enhancements from urban influences. The seasonal cycles of $CO_2$ and $CH_4$ indicate enhancements in the urban sites in our network relative to the more rural stations throughout the year (Fig. 11). Summertime $CH_4$ at urban sites is not as enhanced compared to the rural sites as it is in winter, possibly due to wetland sources influencing the background station at BUC or

lower $CH_4$ emissions from natural gas in urban areas. Similarly, for $CO_2$, some of the rural stations surrounded by active vegetation (Fig. 3) are likely to show stronger influence from biospheric uptake than urban sites in the summer months especially (Fig. 10). We specifically caution against using any of the in-situ data from the NEC rural stations directly as a background for analysis of the urban enhancement without examining these issues. Sargent et al. (2018) indicate that for an analysis of $CO_2$ enhancements in the Boston urban area, $CO_2$ observations from upwind stations alone did not represent the correct background. Even when the air that reaches an urban tower originates near an upwind rural site, back trajectories (from a Lagrangian Particle Dispersion Model such as STILT, for example) indicate that much of the air may originate from a higher altitude than the upwind station. Thus the measurement at an upwind station is not necessarily representative of the proper background, or incoming, concentration, given the large concentration gradients between measurements within the planetary boundary layer and in the free troposphere near background stations with local fluxes. Mueller et al. (2018) conducted an analysis of the issues concerning background determination for this urban network, mostly concerning the large emissions of both $CO_2$ and $CH_4$ upwind of the region that is difficult to capture by upwind stations. We will examine the proper background for investigating urban enhancements in the Washington DC and Baltimore, MD area further in future work.

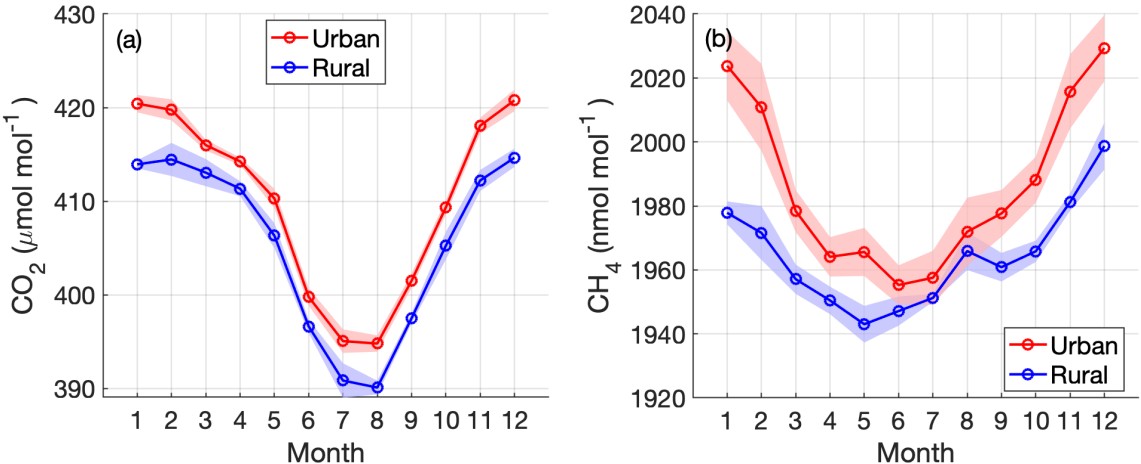

**Figure 11. Seasonal cycles from urban and rural sites in the Washington DC /Baltimore region with at least one year of observations. Mid-afternoon (13-18 LST) daily averages are detrended using a linear fit to the annual trend at Mauna Loa (for $CO_2$) and the global average (for $CH_4$) (data from NOAA/ESRL) and then averaged monthly. Rural sites include TMD, SFD, and BUC; urban sites are ARL, NDC, JES, HAL, NEB, and NWB. Shading indicates one standard deviation of the averages from all the sites.**

## 6 Conclusions

Here we present a data set of hourly average observations of $CO_2$, $CH_4$ and CO (where applicable) from a network of towers in the north-eastern United States. Measurements are funded by NIST and conducted in a collaboration with Earth Networks, Inc., with quality control, assurance, and uncertainty determination conducted by a science team that includes NIST, Earth Networks, and collaborators from the Los Angeles Megacities Carbon Project from NASA/JPL and the Scripps Institution of Oceanography. We present four calendar years of data (2015 through 2018), with different stations coming online through the

years, and most Washington, D.C. and Baltimore, MD urban stations becoming established after late 2015. We also have presented our methodology for calibrating the measurements to WMO scales for each gas and determining uncertainties for

these measurements, as recommended by the WMO (WMO, 2018). We show that analysis of observations at two different inlet heights can be useful for determining the presence of emissions close to the towers, which may be necessary for evaluating the efficacy and choice of transport model used to analyse the data. We also note that the tower stations that were established to characterise incoming or background air are not necessarily appropriate for use directly as background for the urban stations, as they often are affected by local fluxes that do not influence the urban stations. A more careful treatment of incoming

background air is necessary for any given analysis.

## 7 Data availability

This data set of hourly-averaged observations from the Northeast Corridor tower-based network is available on the NIST data portal at data.nist.gov under the DOI 10.18434/M32126 (Karion et al., 2019). Initially, the repository will contain data from 23 sites (Table 1) for years spanning 2015-2018; not all years are available for all sites. Files are version-dated, and the current

plan is to provide annual updates for 2019 and beyond.

## Acknowledgements

We acknowledge the Earth Networks engineering and technical team, including Uran Veseshta, Clayton Fain, Bryan Biggs, Seth Baldelli, Joe Considine, and Charlie Draper. We also thank Tamae Wong, Kimberly Mueller, Sharon Gourdji, Subhomoy

Ghosh, and Antonio Possolo (NIST) for helpful discussions.

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
