# Peer review of "Greenhouse gas observations from the Northeast Corridor tower network"

_Earth System Science Data, 2019_

## Referee Comment (RC1) · Anonymous Referee #1 · 19 Dec 2019

Karion et al. describe a tall tower network for the Washington, D.C., and Baltimore area for measurements of $CO_2$ and $CH_4$. The goal of the network is to provide $CO_2$ and $CH_4$ data to constrain models that quantify GHG emissions from northeastern U.S. metropolitan areas. They further describe a thorough uncertainty analysis for their measurements, especially when limited by the number of calibration standards they are able to use, as well as a future plan for determining uncertainty. Finally, they present some initial data from the network, showing how the vertical gradients in $CO_2$ and $CH_4$ are influenced by some of the sources and sinks surrounding the tower locations.

Overall, this paper provides a good description of the measurement network that will hopefully be used for many years to come. It is certainly worthy of publication in this journal. I only have a few minor comments, listed below.

[Figure]

My only real complaint/disappointment with the paper has to do with the re-calibration of the standard gases by NOAA GMD. I wonder how drift in the calibration standards would affect the uncertainty calculations. However, this is probably a subject for a future paper, not this one.

line 13, I think of the "network" referring to the stations, not the measurements themselves. Perhaps use "observation stations" instead of "observations"?

line 16, suggest "northeast U.S." instead of "US northeast"

line 49, "powerplants" should be two words

Table 1, I'm not sure (mm/yyyy) is necessary, especially since the one digit months do not have a 0 in front. I suggest (mo./yr.)

line 135, suggest "sample" rather than "pull air", and remove "one" from the sentence.

line 141, I assume that's also OD

line 147, this looks to be a Picarro. I'd mention that, since the authors have mentioned the other companies (Valco, Permapure, etc.)

line 190, suggest "12C/13C ratio in CO2" for the general readership

line 278, in the equation "m + b = 1", is that a lower case "L" or a "one"? I assume it's a "one", but it's confusing, because later in the sentence the "SDC,cal = 1" is clearly a non-italicized "one".

line 346, suggest "analyzed" instead of "measured"

line 508 and elsewhere, suggest "by Verhulst. . ." line 560, it seems these numbers should have negative signs to be consistent with Figure 9.
* * *

---

## Referee Comment (RC2) · Anonymous Referee #2 · 21 Jan 2020

The paper presents an interesting dataset which is very important in the contexts of climate negotiations. The paper is well written and presents enough details to allow for evaluation of the quality of the presented dataset.

There are several technical and editorial points to be addressed prior to publication.

lines 28-29: please consider revising the language as "atmospheric measurements" do not detect trends by themselves, rather an analysis of those measurements

lines 47-51: the sentence is confusing, it reads as if agriculture is not included in anthropogenic emissions. Do you actually talk about the separation of fossil fuel and biogenic emissions?

line 75: section 4 presents etc

line 105: are the coordinates of the planned sites known? If so, please include them in the table. Could you please explain why the inlet heights are different. What impact on the station footprint does the change of the inlet height have?

line 113: please spell out "USGS". The intensity in development is used from the 2011 database. Has there been an evaluation of the trend in the intensity since 2011? Can those changes have an impact on the stations' representativeness?

line 131: please spell out "EN GHG"

line 141: what defined 20 min sampling period?

line 155: please spell out "WMO"

line 167: WMO is the World Meteorological Organization

lines 175-179: in the calibration procedure, the first 10 to 15 min of data are filtered out to allow system for equilibration. It seems not to be the case for ambient air sampling. Could you please explain why you cut 75% of data in calibration and do not do the same in the other case (lines 330-340)?

line 231 states that the sensitivity is time-varying, though the calibrations happen every 22 hours. What is the temporal scale of variability you are talking about (as current calibration cycle misses diurnal variability)? What process drives the variability?

line 420: how does this equation account for the uncertainty of the sampling itself?

line 482: are there 1000 physical tanks?

line 544 that states that each measurement period represents a full hour is in contradiction with the earlier elaborations related to the comparison with NOAA flask sampling described in lines 380-387 (where it says that 40 min maybe not representative of the hourly variability)

line 546 refers to the vertical gradient that is calculated based on the measurements

taken at the different heights and probably representing different parts of the planetary boundary layer. How reasonable is putting all estimates in one dataset and elaborate on the physical processes driving variability of the vertical gradients under such an approach?

---

## Referee Comment (RC3) · Anonymous Referee #3 · 2 Feb 2020

This paper presented the greenhouse gas observation network in the organization, structure, instrumentation and measurements of the Northeast Corridor, and gave the greenhouse gas analysis for vertical gradients and seasonal cycles in the urban and rural areas. Besides, this paper shares about 4 years greenhouse gas observation data ($CO_2$, $CH_4$, CO) of 23 sites. Overall, this manuscript is clear and well written. However, following questions are needed to answer:

(1) This paper provides the observation data from 2015 to 2018. I think the short observation time and few observation stations are the weaknesses of this paper. There should be other observation data in the selected urban area in this study. So, I suggest that the author compare with other data and describe the characteristics and innovation points of this data.

(2) What's the differences between single-point calibration and multiple-point calibration? What's the reference data when you did single-point and multiple-point calibration?

(3) Is there any quality flag in the observation data?

———————————————————

---

## Author Comment (AC1) · 10 Feb 2020

Response to Reviewers

We thank the reviewers for their thoughtful and constructive comments. We have addressed each comment below, with the original comment re-stated in italics followed by our response in regular font.

**Reviewer 1**

*Overall, this paper provides a good description of the measurement network that will hopefully be used for many years to come. It is certainly worthy of publication in this journal. I only have a few minor comments, listed below.*

We thank the reviewer for the careful reading and helpful comments.

*My only real complaint/disappointment with the paper has to do with the re-calibration of the standard gases by NOAA GMD. I wonder how drift in the calibration standards would affect the uncertainty calculations. However, this is probably a subject for a future paper, not this one.*

As the reviewer notes, the topic of tank drift was not investigated in this work, as we do not have information yet on how some of the standards may have drifted during the network deployment. We do not expect them to drift significantly (i.e. more than their stated uncertainty) for $CO_2$ or $CH_4$, as prior studies have seen little to no drift in the tanks provided by NOAA (Andrews et al., 2013). We also note that the target tank residual will increase if one of the two tanks drifts relative to the other, and that will be captured in the calculated uncertainty. Only if the target and calibration tanks drift in the same direction would this not be captured. However, for CO, we suspect tank drift may be a factor in our large flask vs. in-situ discrepancy at Mashpee (MSH), where we do see relatively large target tank residuals as well. This issue at MSH is still under investigation.

*line 13, I think of the "network" referring to the stations, not the measurements themselves. Perhaps use "observation stations" instead of "observations"?*

This change has been made, thank you.

*line 16, suggest "northeast U.S." instead of "US northeast"*

Done.

*line 49, "powerplants" should be two words*

Done.

*Table 1, I'm not sure (mm/yyyy) is necessary, especially since the one digit months do not have a 0 in front. I suggest (mo./yr.)*

We agree and have made this suggested change.

*line 135, suggest "sample" rather than "pull air", and remove "one" from the sentence*

We have made the change to "sample" but we actually inserted "and one line samples from a lower level", because that is what we were trying to convey. Two lines sample from the top level. The third line samples from the lower level.

*line 141, I assume that's also OD*

Yes, and OD has been added now.

*line 147, this looks to be a Picarro. I'd mention that, since the authors have mentioned the other companies (Valco, Permapure, etc.)*

Yes, added.

*line 190, suggest "12C/13C ratio in CO2" for the general readership*

Changed.

*line 278, in the equation "m + b = 1", is that a lower case "L" or a "one"? I assume it's a "one", but it's confusing, because later in the sentence the "SDC,cal = 1" is clearly a non-italicized "one".*

We have changed all the "ones" to be not italicized so that it is clearer that they are not "L"'s. Thanks for pointing this inconsistency out.

*line 346, suggest "analyzed" instead of "measured"*
*line 508 and elsewhere, suggest "by Verhulst: : :"*

Both changes have been made.

*line 560, it seems these numbers should have negative signs to be consistent with Figure 9.*

We agree, and this has been done now.

**Reviewer 2**

*The paper presents an interesting dataset which is very important in the contexts of climate negotiations. The paper is well written and presents enough details to allow for evaluation of the quality of the presented dataset.*
*There are several technical and editorial points to be addressed prior to publication.*

We thank the reviewer for the comments and have addressed the points below.

*lines 28-29: please consider revising the language as "atmospheric measurements" do not detect trends by themselves, rather an analysis of those measurements*

That is a good point; the change has been made in the text.

*lines 47-51: the sentence is confusing, it reads as if agriculture is not included in anthropogenic emissions. Do you actually talk about the separation of fossil fuel and biogenic emissions?*

The reviewer makes a good point that we have confused biogenic with natural and included agriculture incorrectly here. The statement has been revised to remove the reference to agricultural sources as separate from anthropogenic sources.

*line 75: section 4 presents etc*

Thank you, this has been fixed ("presents" added)

*line 105: are the coordinates of the planned sites known? If so, please include them in the table. Could you please explain why the inlet heights are different. What impact on the station footprint does the change of the inlet height have?*

The general locations are known, although until lease agreements with the providers are in place we cannot provide exact locations because they may still change. We have updated the locations of the sites in Bluemont, VA, Burke, VA, and Waterford Works, NJ, in both Table 1 and Figure 1, as those site locations are certain and lease agreements on those towers are in place. We have added an explanation in the Table legend as well.

The inlet heights are different due to the towers that were available in desired locations having different heights. We generally install the inlet as close to the top as possible; although we would like all the towers to be at 100 m that is not always feasible. We have done some analysis with a model (WRF-STILT) that indicates that for many sites the footprint does not differ between heights in the middle of the afternoon when the PBL is well-mixed. However, that is not always true, especially if the site is close to sources (which many of them are) or at night. This is indicated by the gradient analysis later in the paper, which shows that the different inlet heights measure slightly different concentrations even in the middle of the afternoon at some urban sites. Thus, we would expect that measurements at 30 meters differ from those at 100 meters, even in the mid-afternoon, depending on the surrounding emissions sources. We do think this topic is important which is why we present the inlet heights in the table along with the brief analysis of vertical gradients later in the manuscript; we hope that users of this data set keep the inlet heights in mind in their analysis.

We have now added the following to the text (107-110): "Although inlet heights were desired to be 100 m above ground level (agl), often shorter towers were used due to the lack of availability of tall towers in ideal locations; the shortest tower in this network has the uppermost inlet at 38 m agl (HRD)."

*line 113: please spell out "USGS".*

Done.

*The intensity in development is used from the 2011database. Has there been an evaluation of the trend in the intensity since 2011? Can those changes have an impact on the stations' representativeness?*

Thank you for this comment. We have updated Figs 2& 3 now using the 2016 NLCD database and found little to no difference between the two years. See the figure below, indicating the fraction for the 4 developed categories for the sites in figure 2 between the two products. The regional sites do not change very much either. We have updated the paper figures to use the 2016 database, as that is indeed more applicable to our data which covers 2015-2018.

[Figure]

*line 131: please spell out "EN GHG"*

These acronyms were both spelled out earlier in the text and are used throughout the manuscript. However, we agree that in this beginning of the section it may be helpful to spell out Earth Networks once more, so we define it again, and remove "GHG" as it is not needed here.

*line 141: what defined 20 min sampling period?*

The 20-minute sampling period was chosen so that each of the three inlets is sampled in each hour. This allows for hourly calculation of differences. Some text has been added to this effect in the document.

*line 155: please spell out "WMO"*
*line 167: WMO is the World Meteorological Organization*

Thank you for catching these errors, they are fixed (WMO defined at first reference, and properly!).

*lines 175-179: in the calibration procedure, the first 10 to 15 min of data are filtered out to allow system for equilibration. It seems not to be the case for ambient air sampling.*
*Could you please explain why you cut 75% of data in calibration and do not do the same in the other case (lines 330-340)?*

We neglected to indicate in the original manuscript that the first 10 minutes of data are indeed filtered when the analyzer switches from a standard (calibration, target, or high standards) to ambient air, but only one minute is filtered when it switches between ambient levels. We made this decision assuming that the concentration difference between ambient levels will be smaller and should equilibrate after a shorter time, while switching from a standard tank to ambient air and vice-versa could be a large difference that requires longer equilibration. The other main reason is the need to flush out the regulators after they remain with stagnant air for 22 hours between samples. We have added text in the first section (was 175-

179, now 205 in the revision): "The first 10 minutes of the ambient air sample following a standard run are also filtered for equilibration, and the first one minute of each 20-minute ambient air run is filtered if it follows another ambient air run (i.e. an inlet switch). The longer flush time is desired for the standard runs because of the need to flush stagnant air remaining in the regulators and tubing when sampling from the cylinder, while the ambient air lines are continuously flushed.".

*line 231 states that the sensitivity is time-varying, though the calibrations happen every 22 hours. What is the temporal scale of variability you are talking about (as current calibration cycle misses diurnal variability)? What process drives the variability?*

Indeed, the sensitivity is time-varying on the scale of the calibration frequency, so at the 22-hour cycle, as we calculate it (the real sensitivity may vary on shorter scales as well). The goal of 22-hour cycle is to avoid sampling the calibrations at the same time every day. The variability of the analyzer response has different frequency characteristics, and the process is unknown. We have found in tests the same variability (1-sigma standard deviation, for example), no matter what the sampling rate. We show Allan Deviation tests for the Picarro analyzers in Verhust et al., ACP, 2017 (SI Figure S8) for the shorter time scales, but we in general find that the variability is the same (in magnitude) at all the temporal scales. Sometimes, there may be longer-term (monthly/annual scale) analyzer drift in the sensitivity that we are trying to capture by using a time-varying sensitivity. We do not know the cause of long-term analyzer drift.

*line 420: how does this equation account for the uncertainty of the sampling itself?*

By sampling, we assume the reviewer is referring to errors that can arise from issues with the sample stream, i.e. the tubing, the valve, etc., either from materials, small leaks, or other unknown effects. Eq. 8 does not account for these effects, which would be largely unknown unfortunately. We hope that they are encompassed in the target tank residual (Eq. 9), which supersedes and is used if it is larger than the uncertainty as calculated using Eq. 8. However, leaks in the sampling line that do not affect the standards (leaks in the tubing before the sample arrives at the calibration box) would not be detected using these variables, and indeed, cannot be detected using the system in an automated way. In practice, these are discovered by monitoring the variability of the air measurements, their absolute CO2 magnitude relative to other similar sites, looking for sudden changes in the concentration time series, or observing the differences between inlets (either measurements from the two inlets at the same height disagreeing systematically, or sudden increase in differences between lower and upper inlets). We also monitor the level differences to ensure that the data from a given inlet is actually being pulled from that inlet (i.e. to discover possible mistakes during installation where tubing gets switched). When leaks are discovered and confirmed with a site visit, all the measurements affected are flagged and removed from the data. The uncertainty calculation does not account for leaks or other sampling issues that are not indicated by the water vapor or target tank data.

We now believe some of these additional quality checks that we perform should be mentioned in the paper, as it might be useful to readers who are operating similar systems. We have included some additional information in Section 3.5, Line 375: "Some additional quality checking is performed at this stage, specifically checking for systematic differences between measurements from two different inlets at the same height and checking for inconsistency in the difference between measurements at different heights. For example, if the lower inlet is systematically reading lower $CO_2$ than the upper inlet, especially at night, it would indicate that the inlet lines may be switched (mislabelled) or there is a leak occurring. These indications would be then verified by a field technician and the data either re-processed or flagged accordingly."

*line 482: are there 1000 physical tanks?*

No, there are only the tanks shown on the figure, i.e. 5 tanks. We did a bootstrap of 1000 random choices of two tanks from the five. We have now clarified this ambiguity in the text by rephrasing.

*line 544 that states that each measurement period represents a full hour is in contradiction with the earlier elaborations related to the comparison with NOAA flask sampling described in lines 380-387 (where it says that 40 min maybe not representative of the hourly variability)*

What we meant in line 544 is that for the gradient analysis, we are making the ***assumption*** that the measurement is representative of the full hour even though the concentrations are not measured through the entire hour. As the reviewer notes, the flask comparison also makes this assumption, and in the discussion of possible reasons for disagreement between the flask and in-situ systems, we note that this assumption may be flawed, causing larger differences than expected between the flask and in-situ systems.

*line 546 refers to the vertical gradient that is calculated based on the measurements taken at the different heights and probably representing different parts of the planetary boundary layer. How reasonable is putting all estimates in one dataset and elaborate on the physical processes driving variability of the vertical gradients under such an approach?*

We thank the reviewer for making a very good point here, in that we have averaged together gradients from sites that are sampling air at different heights. The gradient itself (at night especially) is a function of height – we would expect larger gradients at lower heights, which we do find in our data. We also see larger gradients at sites that have more nearby sources, so these two effects are confounded in this analysis, especially because the towers located in more dense urban areas (NEB, NWB) are also shorter. We have revised Fig. 9(b) to now only include three urban sites that sample at similar levels (lower level at 50 m and upper level at ~90 m), and the background sites we show do also sample at very similar levels (around 50 m and 100 m). This allows for interpreting differences in the gradient as differences in the local fluxes. Now that we show only three urban sites, we are showing each individual trace rather than the mean and deviation as we did in the previous version of this figure. [Also note that due to smaller CH4 gradient in 9(b), the axes were rescaled, and they were rescaled for the following figure with the background site gradients as well for consistency]. We have edited the text accordingly in this section.

**Reviewer 3**

*This paper presented the greenhouse gas observation network in the organization, structure, instrumentation and measurements of the Northeast Corridor, and gave the greenhouse gas analysis for vertical gradients and seasonal cycles in the urban and rural areas. Besides, this paper shares about 4 years greenhouse gas observation data (CO2, CH4, CO) of 23 sites. Overall, this manuscript is clear and well written.*

We thank the reviewer for their reading of the paper and comments, addressed below.

*However, following questions are needed to answer:*

> *(1) This paper provides the observation data from 2015 to 2018. I think the short observation time and few observation stations are the weaknesses of this paper. There should be other observation*

*data in the selected urban area in this study. So, I suggest that the author compare with other data and describe the characteristics and innovation points of this data.*

To our knowledge, there are no other stationary, in-situ, CO2/CH4 concentration measurements being made in ambient air in the Washington DC/Baltimore region. The NOAA/ESRL tall tower network does measure CO2 continuously at 2 sites within our regional domain: a tower in Maine and one in Virginia (Shenandoah National Park), in addition to flask measurements of both CO2 and CH4 at these sites and as mentioned in our paper at MSH and LEW. We have compared with the flask measurements in Section 3.6. We have compared our measurements with the Maine and Virginia towers, as well, but believe such a comparison of observations at different locations and influenced by different surface fluxes is beyond the scope of this paper, which is focused in presenting the data that we have and describing our processing and data quality methods. We do not claim that these are innovative, but are presenting them for transparency to potential users of the data. We also point out that although our record is currently only 4 years, we plan to continue to update the data repository annually, as we intend for this to be a long-term record. We did not believe it was appropriate to wait for a longer record before making the data (and methods) public.

*(2) What's the differences between single-point calibration and multiple-point calibration? What's the reference data when you did single-point and multiple-point calibration?*

A single-point calibration uses only a single known value to derive the calibration curve of the instrument. This value is from sampling a standard tank with known concentration. When a multiple point calibration is used, more than one standard is used, typically one at ambient level (referred to as "calibration standard (cal)" in the text) and one at a higher concentration (referred to as a "high-concentration standard (hstd)" in the text). The text in Sections 3.3 and 3.4 describe the different methodologies in detail, and we have now added a sentence in the beginning of each of these sections defining what we mean by single-point and multiple-point calibration, to clarify this distinction. All of our standards are referenced to World Meteorological Organization scales (L195-196 in the new draft).

*(3) Is there any quality flag in the observation data?*

No, there is not. In lieu of a quality flag, we have opted to provide a numerical uncertainty on each observation, derived as described in the text. We believe this is superior to providing a flag, as it provides the user more information. We do not report data that is of such poor quality that it would not be able to be used in any analysis (for example, during times when a known leak has occurred from the room).

---

## Author Comment (AC2) · 10 Feb 2020

The comment was uploaded in the form of a supplement:
https://www.earth-syst-sci-data-discuss.net/essd-2019-206/essd-2019-206-AC2-supplement.pdf

———————————————————